# Rapidly Mixing Multiple-try Metropolis Algorithms for Model Selection Problems

**Hyunwoong Chang**[*]
Department of Statistics
Texas A&M University
`hwchang@stat.tamu.edu`

**Changwoo J. Lee**[*]
Department of Statistics
Texas A&M University
`c.lee@stat.tamu.edu`

**Zhao Tang Luo**
Department of Statistics
Texas A&M University
`ztluo@stat.tamu.edu`

**Huiyan Sang**
Department of Statistics
Texas A&M University
`huiyan@stat.tamu.edu`

**Quan Zhou**
Department of Statistics
Texas A&M University
`quan@stat.tamu.edu`

## Abstract

The multiple-try Metropolis (MTM) algorithm is an extension of the Metropolis-Hastings (MH) algorithm by selecting the proposed state among multiple trials according to some weight function. Although MTM has gained great popularity owing to its faster empirical convergence and mixing than the standard MH algorithm, its theoretical mixing property is rarely studied in the literature due to its complex proposal scheme. We prove that MTM can achieve a mixing time bound smaller than that of MH by a factor of the number of trials under a general setting applicable to high-dimensional model selection problems with discrete state spaces. Our theoretical results motivate a new class of weight functions called *locally balanced weight functions* and guide the choice of the number of trials, which leads to improved performance over standard MTM algorithms. We support our theoretical results by extensive simulation studies and real data applications with several Bayesian model selection problems.

## 1 Introduction

The Markov chain Monte Carlo (MCMC) method has become a powerful and standard tool for Bayesian posterior computation. In particular, the Metropolis-Hastings (MH) algorithm is widely employed in many statistical and machine learning models to sample from posterior distributions with intractable normalizing constants. A prominent usage example of MH is the vast class of Bayesian model selection problems with discrete-valued high-dimensional parameters, whose posterior probabilities are infeasible to evaluate because the size of model space potentially grows (super-)exponentially with the number of parameters. Examples of Bayesian model selection problems and their applications are ubiquitous, including Bayesian variable selection (BVS) [45, and references therein], stochastic block model (SBM) [20, 34], Bayesian structure learning [10, 15, 29], change-point detection model [5, 11], spatial clustering models [23, 26, 28], and many others.

Despite the popularity of MCMC algorithms, their convergence rate analysis is often challenging. Mixing time is a key concept of interest when analyzing the convergence of MCMC, which defines the number of iterations needed to converge to a stationary distribution within a small total variation distance [25]. We say a chain is *rapidly mixing* if the mixing time grows at most polynomially in some complexity parameters. Mixing time bound has been studied extensively for problems such as card shuffling [3], random walks on groups [2], graph colorings [21], and the Ising model [14], across

---

[*]Equal contribution

36th Conference on Neural Information Processing Systems (NeurIPS 2022).

the fields of probability, physics, and computer science [40]. Although there is a rich literature on discrete-state-space mixing time analysis in many areas, the focus has been primarily on approximate counting and random generation, but only recently has research on the MCMC convergence in the context of statistical model selection begun to emerge. [47] proved rapid mixing of an MH algorithm in a high-dimensional BVS problem, and [38] improved the bound when a "warm start" is available. Rapid mixing results of MH algorithms are established for Bayesian community detection problem with SBM [55] and for learning equivalence class in a high-dimensional Bayesian structure learning [52]. Recently, locally informed MH algorithms have been developed to incorporate local information about the target distribution. [48] proposed a locally balanced proposal and proved their optimality in terms of Peskun ordering [37] for discrete space models without providing mixing time rate analysis. [54] also proposed an informed MCMC and proved a dimension-free mixing time bound but only for a BVS problem.

The multiple-try Metropolis (MTM) algorithm [27] is an extension of the MH algorithm, which uses a weight function to choose the proposed state among randomly sampled trials from a proposal distribution. The variety of weight functions and the number of trials enrich the algorithm class but result in a much more complex transition probability than standard MH algorithms, hampering the theoretical development on the mixing time of MTM. As a result, there is a lack of theoretical guidance for practitioners on the choices of weight functions and number of trials in MTM and also the choice between MTM and other MCMC algorithms. Only very recently, [46] considered the MTM independence sampler (MTM-IS), a special case of MTM where the proposal does not depend on the current state. They conducted the convergence rate analysis of MTM-IS, which shows it is less efficient than the simpler approach of repeated Metropolised independent sampling at the same computational cost, but they did not discuss the choices of weight functions and the number of trials. As acknowledged in [46], MTM is significantly different from MTM-IS, and its convergence rate is more challenging to analyze. In this paper, we establish the mixing time bound of MTM for general model selection problems. Our main contributions are the following:

1) We show that MTM with existing popular choices of weight functions can have mixing issues and propose a new class of weight functions called *locally balanced weight functions*.

2) With locally balanced weight functions, we prove that the mixing time bound of the MTM algorithm is smaller than that of the MH algorithm by a factor of the number of trials $N$ in a general model selection setting, under some regularity conditions including a rate condition on $N$.

3) We provide theoretical justification for the counterintuitive phenomenon that an increase in the number of trials $N$ may not always lead to a proportionate improvement in mixing, and suggest a theoretically guided algorithm to choose $N$ for practical use.

4) We validate our theoretical findings via extensive simulation studies and real data applications with various model selection problems: Bayesian variable selection, stochastic block models, spatial clustering models, and structure learning models.

## 2 Preliminary

### 2.1 Notation

For a positive integer $m \in \mathbb{N}$, let $[m] = \{1, \ldots, m\}$ and $\lfloor m \rfloor$ be the largest integer no greater than $m$. Let $|\cdot|$ denote the cardinality of a set. Let $\mathcal{X} = \mathcal{X}_p$ denote a finite state space where $|\mathcal{X}|$ grows in a complexity parameter $p \in \mathbb{N}$. We use $\mathbf{P} : \mathcal{X} \times \mathcal{X} \to [0, 1]$ to denote the transition probability matrix of an irreducible, aperiodic and reversible Markov chain. We denote by $\mathbf{I}$ an identity matrix. We call a probability distribution $\pi$ on $\mathcal{X}$ a *stationary distribution* or a *target distribution* of a chain $\mathbf{P}$ if it satisfies $\sum_{z \in \mathcal{X}} \pi(z)\mathbf{P}(z, x) = \pi(x)$ for all $x \in \mathcal{X}$. In Bayesian inference, $\pi$ is a posterior distribution. Let $\mathcal{N} \colon \mathcal{X} \to 2^{\mathcal{X}}$ be a set-valued map called a neighborhood relation, which maps a state to a set of states. We say the neighborhood relation $\mathcal{N}$ is symmetric if $x' \in \mathcal{N}(x)$ iff $x \in \mathcal{N}(x')$ for all $x \neq x'$. We define the random walk proposal matrix as $\mathbf{K}_{\mathrm{RW}}(x, y) = \mathbb{1}_{\mathcal{N}(x)}(y)/|\mathcal{N}(x)|$ where $\mathbb{1}_{\mathcal{N}(x)}$ denotes the indicator function of $\mathcal{N}(x)$. We say $\mathbf{K}$ is symmetric if $\mathbf{K}(x, y) = \mathbf{K}(y, x)$. We denote a graph by $(V, E)$ where $V$ is the vertex set and $E \subset V \times V$ is the set of edges. We say $\delta(x, y) = (v_0, \ldots, v_m)$ is an $E$-path (with length $m$) from $x$ to $y$ if $e_i = (v_{i-1}, v_i) \in E$ for $i \in [m]$, $v_0 = x, v_m = y$, and all vertices are distinct.

---

**Algorithm 1:** Multiple-try Metropolis (MTM) algorithm with proposal $\mathbf{K}_{\mathrm{RW}}$

---

**Input:** An initial state $x_0$, a neighborhood relation $\mathcal{N}$, the number of trials $N$, a weight function $w$ defined in (1), the number of Markov chain iterations $T$.

1 **for** $t = 0, \ldots, T - 1$ **do**

2     Step 1. Draw $y_1, \ldots, y_N$ uniformly at random from $\mathcal{N}(x_t)$, and compute $w(y_j \mid x_t)$ for $j = 1 \ldots, N$.

3     Step 2. Select $j \in [N]$ with probability proportional to $w(y_j \mid x_t)$ and define $y = y_j$.

4     Step 3. Sample $x_1^\star, \ldots, x_{N-1}^\star$ uniformly at random from $\mathcal{N}(y)$ and define

$$\alpha = \min \left\{ 1, \frac{w(y \mid x_t) + \sum_{l \in [N] \setminus \{j\}} w(y_l \mid x_t)}{w(x_t \mid y) + \sum_{l \in [N-1]} w(x_l^\star \mid y)} \right\}. \tag{2}$$

5     Step 4. With probability $\alpha$, accept $y$ and let $x_{t+1} = y$; otherwise, let $x_{t+1} = x_t$.

**Output:** A set of Markov chain samples $\{x_t\}_{t=1}^T$.

---

### 2.2 Multiple-try Metropolis algorithm

We present the MTM algorithm [27] on a discrete state space in Algorithm 1. Here $N$ trial states $y_1, \ldots, y_N$ are first sampled from the proposal $\mathbf{K}_{\mathrm{RW}}(x, \cdot)$, and then the proposed state $y$ is selected among the trials using the weight function

$$w(y \mid x) = \pi(y) \mathbf{K}_{\mathrm{RW}}(y, x) \lambda(y, x), \tag{1}$$

where $\lambda(y, x)$ can be any symmetric function that is positive whenever $\mathbf{K}_{\mathrm{RW}}(y, x) > 0$. Note that Algorithm 1 becomes the standard random walk MH algorithm when the number of trials $N$ is one.

There have been efforts to generalize the MTM algorithm [9, 12, 30]. [35] extends the weight function class by proposing a general form of acceptance probability to achieve the reversibility of a chain when the weight function does not belong to (1). While one can easily find a weight function satisfying (1), there is little research on the most appropriate form of weight functions. [27] investigated some particular class of weight functions and claimed the performance is insensitive to the choice of weight functions within this class; $w(y \mid x) = \pi(y)$ (when $\mathbf{K}_{\mathrm{RW}}$ is symmetric) and $w(y \mid x) = \pi(y)/\mathbf{K}_{\mathrm{RW}}(x, y)$ are popular choices in this class [6, 33].

Another important component of the MTM method is the number of trials $N$. It has been reported with empirical evidence that increasing $N$ does not necessarily result in a corresponding improvement in mixing under the random walk proposal [32]. They illustrate a counterintuitive phenomenon that the acceptance probability for a proposal move to higher probability regions can be extremely low. As a solution, they suggest randomly selecting $N$ at each iteration. Despite the anecdotal observation that certain choices of $N$ make the algorithm less efficient, the principled way to choose $N$ is rarely discussed in the literature.

### 2.3 Analysis on mixing time via geometric tools

A theoretical foundation for employing MCMC methods to sample from a posterior distribution on a finite state space is the well-known convergence theorem, which states that an irreducible and aperiodic Markov chain with a stationary distribution converges to the stationary distribution regardless of the initial distribution [25, Theorem 4.9]. However, since it is impossible to simulate a chain infinitely long, our main interest is to analyze how well the chain approximates the target distribution after a certain number of steps. To this end, we introduce $\epsilon$-mixing time $t_{\mathrm{mix}}(\epsilon) = \max_{x \in \mathcal{X}} \min \left\{ t \in \mathbb{N} \colon \| \mathbf{P}_{\mathrm{lazy}}^t(x, \cdot) - \pi(\cdot) \|_{\mathrm{TV}} \leq \epsilon \right\}$, where $\| \cdot \|_{\mathrm{TV}}$ is the total variation (TV) distance, $\epsilon$ is a small positive constant, and $\mathbf{P}_{\mathrm{lazy}} = (\mathbf{P} + \mathbf{I})/2$ denotes the lazy version of $\mathbf{P}$, which is introduced merely for theoretical convenience since all eigenvalues of $\mathbf{P}_{\mathrm{lazy}}$ are non-negative (the use of lazy Markov chains is standard in mixing time analysis). In words, $t_{\mathrm{mix}}$ quantifies the necessary number of steps for the chain to have a small TV distance from the stationary distribution. Among several techniques for characterizing the mixing property [8, 18, 25, 42], a path method is employed to derive our main result [13, 43]. We also refer readers to [25, Section 13.4] and [42, Section 3] for more details about the path method. Intuitively, the mixing time depends on how well the states "communicate", especially among those which have high values of $\pi$. In contrast to

samplers defined on a continuous state space with a Euclidean topology, posterior landscape and modality on a finite state space are more difficult to envision since it highly depends on how one defines the neighborhood relation. Given a transition probability matrix $\mathbf{P}$, we define the neighborhood of a state $x$ by $\mathcal{N}(x) = \{x' \in \mathcal{X} \setminus \{x\} : \mathbf{P}(x, x') > 0\}$, the set of reachable states by one-step transition from $x$, and we say $x$ is a mode if $\pi(x) > \max_{x' \in \mathcal{N}(x)} \pi(x')$. This specification of the neighborhood enables us to define a graph $(\mathcal{X}, E)$ where $E = \{(x, x') \in \mathcal{X} \times \mathcal{X} : x' \in \mathcal{N}(x)\}$ is an (undirected since we have a symmetric neighborhood relation by the reversibility of $\mathbf{P}$) edge set. For each $(x, y) \in \mathcal{X} \times \mathcal{X}$ with $x \neq y$, we choose only one $E$-path from $x$ to $y$ and denote it as $\delta(x, y)$. A *path ensemble* is defined as a collection of the paths $\Delta = \{\delta(x, y) : x, y \in \mathcal{X}, x \neq y\}$. A path ensemble is a "configuration of path network" that helps quantify the quality of the communication within the state space $\mathcal{X}$. The following proposition, which serves as a base of our main result, gives a mixing time bound by means of a path ensemble.

**Proposition 1** ([43])**.** *For any path ensemble $\Delta$,*

$$t_{\mathrm{mix}}(\epsilon) \leq 2\rho(\Delta)\ell(\Delta) \left[ \log(1/\epsilon) + \log \left\{ \min_{x \in \mathcal{X}} \pi(x) \right\}^{-1} \right],$$

*where $\ell(\Delta) = \max_{x,y} |\delta(x, y)|$ and*

$$\rho(\Delta) = \max_{(u,v) \in E} \frac{1}{\pi(u)\mathbf{P}(u, v)} \sum_{x,y : \delta(x,y) \ni (u,v)} \pi(x)\pi(y). \tag{3}$$

For completeness, we provide a proof of Proposition 1 with a detailed discussion in Appendix A.1. The quantity $\rho(\Delta)$ is called the *congestion parameter*, and $\ell(\Delta)$ is the length of the longest path in $\Delta$. To interpret $\rho(\Delta)$ and $\ell(\Delta)$, some terminologies from transportation networks in graph theory are often useful. Consider $\pi(u)\mathbf{P}(u, v)$ as the capacity of an edge $(u, v)$, and $\pi(x)\pi(y)$ as the unit flow of a path $\delta(x, y)$. Then $\rho(\Delta)$ quantifies the maximum loading of an edge under the configuration $\Delta$. If one edge with a small capacity is traversed by a huge number of paths with a high unit flow, it results in a constriction. Consider $\ell(\Delta)$ as the diameter of the configuration $\Delta$. If the diameter is extremely large, it causes inefficient communication, especially among the states lying on the margin of the configuration. If it is possible to design the path system in a way that no edge is overloaded by paths and no two states are too far apart, then the chain can have a good mixing behavior. One of the key steps in the proof of the main result is to identify a good path ensemble $\Delta^*$ (see Appendix A.3).

## 3   Main result

In this section, we prove the rapid mixing of the MTM algorithm for general model selection problems on a finite model space $\mathcal{X}$, where we assume that there is only one state with the highest posterior mass, denoted by $x^*$. Throughout the section, we may assume that $|\mathcal{X}|$ grows with the complexity parameter $p$ (e.g., the number of variables in BVS), and the posterior distribution $\pi$ and other related objects are implicitly parameterized by $p$.

### 3.1   A mixing time bound with locally balanced weight functions

While extending the possible form of weight functions has been the main focus in the literature [33, 35], the choice of weight function is rarely discussed. We introduce a class of weight functions by using a function $h \colon \mathbb{R}_{>0} \to \mathbb{R}_{>0}$ that satisfies $h(u) = uh(1/u)$ for all $u > 0$; such a function $h$ is called a *balancing function* [31, 48]. The class of balancing functions is very broad: we can choose an arbitrary non-negative function $h$ on $(0, 1]$, and then $h(u)$ on $u \in (1, \infty)$ is defined by the "balancing rule", $h(u) = uh(1/u)$. Typical choices of balancing functions include $\sqrt{u}$, $\min\{1, u\}$, $\max\{1, u\}$, $u/(u + 1)$, and $u + 1$. The following proposition states that Algorithm 1 with the proposed weight function induces a reversible Markov chain with $\pi$ as its stationary distribution.

**Proposition 2** (Locally balanced weight functions)**.** *Suppose that a weight function is defined as*

$$w(y \,|\, x) = h\left( \frac{\pi(y)\mathbf{K}_{\mathrm{RW}}(y, x)}{\pi(x)\mathbf{K}_{\mathrm{RW}}(x, y)} \right), \tag{4}$$

*where $h$ is a balancing function. Then, a Markov chain $\mathbf{P}_{\mathrm{MTM}}$ induced by Algorithm 1 has a stationary distribution $\pi$.*

The proof is deferred to Appendix A.2. We call weight functions of the form (4) *locally balanced weight functions*; though we only consider random walk proposals in this paper, $\mathbf{K}_{\mathrm{RW}}$ in Proposition 2 can be replaced by any other proposal matrix. Compared with the popular choice of weight function in the existing MTM literature such as $w(y \,|\, x) = \pi(y)$, the locally balanced weight function can be viewed as a transformed and scaled version with respect to the posterior probability at current state $x$. For example, if $h(u) = \sqrt{u}$ and $\mathbf{K}_{\mathrm{RW}}$ is symmetric, then $w(y \,|\, x) = \sqrt{\pi(y)/\pi(x)}$.

Before providing a rigorous mixing time bound of MTM under our new choice of $w(y \,|\, x)$, we give an intuitive explanation for why the existing unscaled weight function may have poor mixing properties. First, it is reasonable to assume that the true data-generating model, $x^*$, will have the largest posterior probability when the posterior contraction or the model selection consistency holds asymptotically. Indeed, such results have been established for a variety of model selection problems [47, 52, 55]. On the other hand, a model space $\mathcal{X}$ is often endowed with a natural neighborhood relation such that the degree of "model fit" (measured by goodness of fit and penalization on model size) between neighboring models tends to have a small difference. This implies that a small modification of a model may lead to a marginal change in the posterior probability. For example, in a Bayesian variable selection problem with the number of variables $p$, the model space can be represented as a set of binary vectors $\gamma \in \{0,1\}^p$ where $\gamma_j = 1$ (resp. $\gamma_j = 0$) indicates $j$-th variable is included (resp. excluded) in the model. Let us assume that $\gamma^*$ is the true data generating model and has the highest posterior probability, and define a Hamming distance $d_{\mathrm{H}}(\gamma, \gamma') = \sum_{j=1}^{p} \mathbb{1}\{\gamma_j \neq \gamma'_j\}$ for $\gamma, \gamma' \in \{0,1\}^p$.

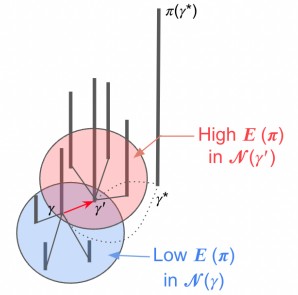

One often defines the neighborhood $\mathcal{N}(\gamma)$ as the set of states with a small Hamming distance to $\gamma$ [19, 47, 54], and models in the neighborhood tend to have similar posterior probabilities. A state with a large Hamming distance from $\gamma^*$ usually has a small posterior probability because its structure is largely different from the true data-generating model. Figure 1 depicts such a relationship between posterior probabilities and structural similarities among the models. Given that $\gamma$ is the current state, Step 2 of Algorithm 1 tends to propose a neighboring state $\gamma'$ with a higher posterior probability, so $\gamma'$ tends to be closer to $\gamma^*$ in Hamming distance, i.e. $d_{\mathrm{H}}(\gamma', \gamma^*) < d_{\mathrm{H}}(\gamma, \gamma^*)$. On average, the Hamming distance between a neighboring state of $\gamma'$ and $\gamma^*$ will be smaller than that between a neighboring state of $\gamma$ with $\gamma^*$. Therefore, the expected value of posterior probability of a neighboring state of $\gamma'$ (in the denominator part in (2), shown in red in Figure 1) will be typically larger than that of a neighboring state of $\gamma$ (in the numerator part in (2), shown in blue in Figure 1). If we resort to the unscaled weight function $w(\gamma' \,|\, \gamma) = \pi(\gamma')$, this can cause a significantly low acceptance probability, especially when the number of trials $N$ is large (due to the law of large numbers, the acceptance ratio converges to the ratio of average posterior probabilities of the two neighborhoods); see also Appendix A.4 for a concrete example. A locally balanced weight function (4), however, is able to mitigate this counterintuitive phenomenon by "scaling" the weight function with respect to the current state. We shall demonstrate the effectiveness

Figure 1: An illustration of the reason that the weight function $w(y \,|\, x) = \pi(y)$ may fail. Each vertical bar represents the posterior probability on the corresponding state, and thin lines connect neighboring states. The expected value of the posterior probability of the neighborhood of $\gamma'$ (red) is typically greater than that of $\gamma$ (blue).

of locally balanced weight functions in the simulation studies. Although the scope of our study is limited to finite state spaces, employing (4) also improves the algorithm on continuous state spaces. We provide the relevant discussion in Appendix D.2.

In order for the MTM procedure to efficiently sample states from a high posterior region, it is sensible to assign a larger weight to those states with a higher posterior probability in the proposal, i.e. Step 2 of Algorithm 1. To this end, we only require $h$ to be non-decreasing. Now we introduce the main result, which states that the mixing time bound can be improved linearly in the number of trials $N$ for our locally balanced weighting scheme.

**Theorem 1.** *Let* $\mathbf{P}_{\mathrm{MTM}}$ *denote a Markov chain induced by Algorithm 1 with a locally balanced weight function (4) with some non-decreasing balancing function h. Define*

$$\mathcal{S}(x) = \{x' \in \mathcal{N}(x) \colon \pi(x')/\pi(x) > p^{t_1}\}, \tag{5}$$

*for some constant $t_1$ and $s_0 = \max_{x \in \mathcal{X}} |\mathcal{S}(x)|$. Suppose that the following conditions hold with $p \geq 2$ and $t_2, t_3, t_4 \geq 0$ being some constants that satisfy $t_1 < t_2$, $t_3 \leq t_4 < t_2$.*

*(i) There exists $x' \in \mathcal{N}(x)$ such that $\pi(x')/\pi(x) \geq p^{t_2}$ for every $x \neq x^*$.*

*(ii) For any $x \in \mathcal{X}$, $p^{t_3} \leq |\mathcal{N}(x)| \leq p^{t_4}$.*

*(iii) $N = o\left(\min\left\{\frac{h(p^{t_2+t_3-t_4})}{h(p^{t_1+t_4-t_3})}, \frac{p^{t_3}}{s_0}\right\}\right).$*

*Then, we have the $\epsilon$-mixing time of $\mathbf{P}_{\mathrm{MTM}}$*

$$t_{\mathrm{mix}}(\epsilon) = O\left(\frac{1}{N}(p^{-t_4} - p^{-t_2})^{-1}\ell(\Delta^*)\log\left(\min_{x\in\mathcal{X}}\pi(x)\right)^{-1}\right), \tag{6}$$

*where $\Delta^*$ is a path ensemble defined in Appendix A.3.*

*Remark* 1. By the recent result of [53, Lemma 3] based on a refined path argument, we can remove the term $\ell(\Delta^*)$ from (6) without changing the order of the bound.

*Sketch of the proof.* The complete proof is deferred to Appendix A.3. The first key step of our proof is to identify a suitable path ensemble $\Delta^*$ by defining a function $g : \mathcal{X} \to \mathcal{X}$ such that $g(x)$ has the highest posterior probability in the neighborhood of the state $x$. With $\pi$ being unimodal and $\mathcal{X}$ being finite, repeated composition of $g$ for any $x$ will always lead to the mode $x^*$, that is $g^m(x) = x^*$ for some $m \in \mathbb{N}$, which we utilize to construct $\delta^*(x, y)$ for any $x, y \in \mathcal{X}, x \neq y$. See Appendix A.5 for an toy example. The other key step is to bound the congestion parameter $\rho(\Delta^*)$ defined in (3). To this end, we prove that the lower bound of the transition probability $\mathbf{P}(x, g(x))$ gets closer to $Np^{-t_4}$ asymptotically, for any state $x \in \mathcal{X}\backslash\{x^*\}$.

Condition (i) implies that the posterior distribution is unimodal with the peak at the model $x^*$. We emphasize that the unimodality is with respect to the neighborhood relation $\mathcal{N}$ and is usually satisfied with some appropriate choice of a large enough neighborhood. For example, for Bayesian variable selection, it is shown in [47] that their proposed posterior may not be unimodal if a neighborhood is defined as a set of models within 1-Hamming distance, but expanding to the 2-Hamming distance neighborhood satisfies Condition (i) with $t_2 = 2$. Similarly, the community detection problem considered in [55] and the structure learning problem in [52] satisfy (i) with some positive constant $t_2$ by choosing an appropriate $\mathcal{N}$. Next, Condition (ii) states that the neighborhood size $|\mathcal{N}(x)|$ should neither be too huge nor vary much from state to state. The purpose of this condition is to control the ratio $\mathbf{K}_{\mathrm{RW}}(y, x)/\mathbf{K}_{\mathrm{RW}}(x, y)$ in (4) so that the posterior ratio $\pi(y)/\pi(x)$ dominates the term. Last but not least, Condition (iii) implies that the number of trials $N$ should not be arbitrarily large. This is consistent with the empirical observation of [32] that additional trials do not always result in better mixing. Condition (iii) also implies that $s_0 = o(p^{t_3})$ since $N \geq 1$. This seems to be a strong condition, since the number of neighboring states with a relatively large posterior probability with respect to the current state can be very large if we consider a state with the smallest posterior that is surrounded by states with greater posterior. Yet, it is a common practice to restrict the support of the prior to rule out unrealistic models by imposing some high-dimensional regularity assumptions, such as sparsity. For example, [47] introduced the parameter of the maximum number of important covariates in variable selection, and [52] considered the maximum in-degree and out-degree in structure learning to restrict the model space. In the same context, [55] suggested the use of a feasible set for the initial partition in the community detection problem. In this regard, we may consider $\mathcal{X}$ as the restricted space so that $s_0$ can be controlled. Most importantly, Condition (iii) provides a key idea on how to select the number of trials $N$, which will be discussed in Section 3.2.

*Remark* 2 (Problem-specific applications). Theorem 1 can be used to prove rapid mixing of the MTM algorithm for Bayesian variable selection [47] and structure learning [52] with order $N$ improvement of the mixing bound, given that the choice of $N$ satisfies the condition (iii).

*Remark* 3 (Computational benefits). Since MTM with $N$ trials requires calculating $2N - 1$ weight functions at each iteration, improving the mixing time bound by the factor of $N$ leads to the same overall computational complexity as the single-try MH algorithm until the convergence of the Markov chain. However, compared to the single-try MH with $N$ times longer sequential iterations, the calculation of $N$ weight functions can be efficiently done in parallel. For example, calculating multiple weight functions can often be converted into a series of matrix multiplication problems, and optimized linear algebra libraries such as BLAS [7] can be used to exploit multiple execution units and pipe-lining. As examples, we outline parallelization strategies for the Bayesian variable selection (Section 4.1) and stochastic block model (Section 4.2) in Appendix D.1. Indeed, Table 1 in our simulation study results shows that MTM with moderate choice of $N$ significantly reduces the wall-clock time until the convergence of the Markov chain.

## 3.2 Choice of the number of trials

Motivated by Condition (iii) in Theorem 1, we propose an algorithm to choose the number of trials $N$ under a general setting applicable to high-dimensional model selection problems. For conciseness, we shall only focus on illustrating the case when we have balancing function $h(u) = \sqrt{u}$ and symmetric proposals (known $t_3 = t_4$) so that Condition (iii) becomes $N = o\left(\min\{p^{(t_2-t_1)/2}, p^{t_3}/s_0\}\right)$, although the algorithm can be easily generalized to other choices of $h$ and non-symmetric proposals.

---

**Algorithm 2:** The choice of the number of trials, when $h(u) = \sqrt{u}$ and known $t_3 = t_4$.

**Input:** An initial state $x_0$, neighborhood $\mathcal{N}$, constant $\psi \in (0,1)$.
1 Step 1. Calculate $\log_p(\pi(y_j)/\pi(x_0))$ for all $y_j \in \mathcal{N}(x_0)$, $j = 1, \ldots, |\mathcal{N}(x_0)|$.
2 Step 2. Run k-means algorithm with k=2 on log-ratios $\{\log_p(\pi(y_j)/\pi(x_0))\}$ to obtain a partition, $\mathcal{C}_1$ (set of small log-ratios) and $\mathcal{C}_2$ (set of big log-ratios).
3 Step 3. Let $\hat{t}_2 = \min \mathcal{C}_2$, $\hat{t}_1 = \max \mathcal{C}_1$ and $\hat{s}_0 = |\mathcal{C}_2|$. [†]
4 **while** $p^{(\hat{t}_2-\hat{t}_1)/2} < p^{t_3}/\hat{s}_0$ **do**
5     Update $\mathcal{C}_1 \leftarrow \mathcal{C}_1\backslash\{\hat{t}_1\}$, $\hat{t}_1 \leftarrow \max \mathcal{C}_1$ and $\hat{s}_0 \leftarrow \hat{s}_0 + 1$.

**Output:** The number of trials $N = \lfloor (p^{t_3}/\hat{s}_0)^\psi \rfloor$.

---

[†] If $\hat{t}_2 < t_4$, redefine $\mathcal{C}_1$, $\mathcal{C}_2$ such that $\hat{t}_1 = \max \mathcal{C}_1 \leq t_4 < \min \mathcal{C}_2 = \hat{t}_2$. If no such $\mathcal{C}_2$ exists, set $\hat{t}_2 = t_4, \hat{s}_0 = 1$ and $\hat{t}_1 = \max \mathcal{C}_1$.

Algorithm 2 is depicted in Figure 2. In Step 1, the log probability ratio is calculated for all neighboring states of the initial state $x_0$. Step 2 identifies two subsets $\mathcal{C}_1$ and $\mathcal{C}_2$, corresponding to "bad moves" and "good moves" respectively. In Step 3, motivated by Condition (i), we set $\hat{t}_2 = \min \mathcal{C}_2$, a lower bound of log probability ratios among good moves. After $t_2$ is fixed, we first set $\hat{t}_1 = \max \mathcal{C}_1$, $\hat{s}_0 = |\mathcal{C}_2|$ and gradually decrease $\hat{t}_1$ (so that $\hat{s}_0$ increases) until the "crossing" $p^{(\hat{t}_2-\hat{t}_1)/2} \geq p^{t_3}/\hat{s}_0$ happens. The choice of $N = \lfloor (p^{t_3}/\hat{s}_0)^\psi \rfloor$ immediately after the crossing point approximates the worst-case scenario of $\min\{p^{(t_2-t_1)/2}, p^{t_3}/s_0\}$, hence providing a conservative estimate of $N$. We set $\psi = 0.9$ to match with the asymptotic dominance condition (iii).

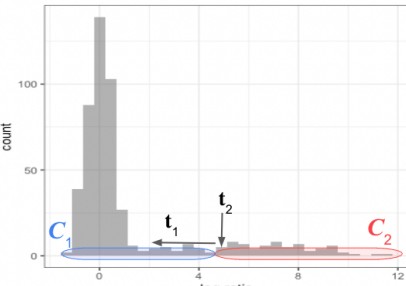

Figure 2: An illustration of Algorithm 2.

We run Algorithm 2 only once for an initial state $x_0$. It is based on the insight that for model selection problems, log probability ratios can often be partitioned into "bad moves" and "good moves". Although we considered any $x \in \mathcal{X}\backslash\{x^*\}$ in Condition (i), analyzing the initial state $x_0$ is not only computationally simple but also gives a good guess of $t_2$ since as chain proceeds to the highest posterior state $x^*$, the number of good moves and the magnitude of ratios generally decreases, for example in the Bayesian variable selection problem [54]. Another possible way of using Algorithm 2 is to re-evaluate $N$ after a certain number of MCMC iterations. We note that another clustering algorithm can be substituted for the k-means algorithm in Step 2.

## 4 Simulation studies

In this section, we present the results for two Bayesian model selection problems, BVS and SBM. We defer the result for spatial clustering models to Appendix B.3 due to the page limit. Throughout the simulation studies, we use symmetric proposal schemes. We run single-try MH and MTM with different choices of $N \in \{5, 10, 50, 100, 500, 1000, 2000, 5000\}$ and four different choices of weight functions, one with (unscaled) ordinary weight function $w_{\text{ord}}(y \mid x) = \pi(y)$, and three with the proposed locally balanced weight functions $w_{\text{sqrt}}(y \mid x) = \sqrt{\pi(y)/\pi(x)}$, $w_{\min}(y \mid x) = \min\{1, \pi(y)/\pi(x)\}$, and $w_{\max}(y \mid x) = \max\{1, \pi(y)/\pi(x)\}$. We consider the scenarios where the data-generating model $x^*$ receives the highest posterior probability to verify our main theoretical results. Similarly to [54], two performance measures are considered: 1) $H = \min\{t : x_t = x^*\}$ (hitting iteration): the number of MCMC iterations until the Markov chain $(x_0, x_1, \ldots)$ hits $x^*$, 2) $T_H$ (wall-clock hitting time): the wall-clock time taken until the chain hits $x^*$.

## 4.1 Bayesian variable selection (BVS)

Consider a high-dimensional linear model with response vector $\boldsymbol{y} \in \mathbb{R}^n$ and design matrix $\boldsymbol{X} \in \mathbb{R}^{n \times p}$, where the number of predictors $p$ is much larger than the sample size $n$. BVS seeks to find the best subset of predictors, denoted as a binary vector $\gamma \in \{0, 1\}^p$ described in Section 3.1, using the sparsity-inducing prior. We adopt the prior introduced by [47]:

$$\text{Linear model:} \qquad \boldsymbol{y} = \boldsymbol{X}_\gamma \boldsymbol{\beta}_\gamma + \boldsymbol{\epsilon}, \quad \boldsymbol{\epsilon} \sim \mathsf{N}\left(\boldsymbol{0}, \phi^{-1}\mathbf{I}_n\right)$$

$$\text{Prior on } \boldsymbol{\beta}_\gamma \text{ and } \phi: \qquad \boldsymbol{\beta}_\gamma \mid \phi, \gamma \sim \mathsf{N}\left(\boldsymbol{0}, \mathscr{G}\phi^{-1}(\boldsymbol{X}_\gamma^\top \boldsymbol{X}_\gamma)^{-1}\right), \quad \pi(\phi) \propto 1/\phi$$

$$\text{Sparsity prior:} \qquad \pi(\gamma) \propto p^{-\kappa|\gamma|}\mathbb{1}\left[|\gamma| \le s_{max}\right],$$

where $\boldsymbol{X}_\gamma$ is a submatrix of $\boldsymbol{X}$ consisting of all $j$-th columns with $\gamma_j = 1$, $\boldsymbol{\beta}_\gamma \in \mathbb{R}^{|\gamma|}$ is the subvector of $\boldsymbol{\beta} \in \mathbb{R}^p$ with nonzero coefficients, where $|\gamma| = \sum_{j=1}^p \gamma_j$. The hyperparameters $\mathscr{G}$, $\kappa$ and $s_{\max}$ control the weight of prior effect [50], sparsity strength and the maximum model size, respectively. We sample $\gamma$ from the posterior distribution $\pi(\gamma \mid \boldsymbol{y})$ where other parameters $\boldsymbol{\beta}_\gamma$ and $\phi$ are marginalized out; see Appendix B.1 for details. We consider the following random walk proposal:

$$\mathbf{K}_{\mathrm{RW}}(\gamma, \gamma') = \begin{cases} \frac{1}{p}\mathbb{1}_{\mathcal{N}_1(\gamma)}(\gamma') \text{ (single flip)} & \text{if } |\gamma| < s_{\max}, \\ \frac{1}{2p}\mathbb{1}_{\mathcal{N}_1(\gamma)}(\gamma') + \frac{1}{2|\gamma|(p-|\gamma|)}\mathbb{1}_{\mathcal{N}_2(\gamma)}(\gamma') \text{ (single or double flip)} & \text{if } |\gamma| = s_{\max}, \end{cases}$$

where $\mathcal{N}_1(\gamma) = \{\gamma' : d_{\mathrm{H}}(\gamma, \gamma') = 1\}$ and $\mathcal{N}_2(\gamma) = \{\gamma' = (\gamma \cup \{j\})\setminus\{\ell\} : j \notin \gamma, \ell \in \gamma\}$. With $n = 1000$ and $p = 5000$, we consider different settings of design matrix where each row $\boldsymbol{x}_i$ is i.i.d. sampled from $\boldsymbol{x}_i \sim \mathsf{N}(\boldsymbol{0}, \mathbf{I}_p)$ (independent) or $\boldsymbol{x}_i \sim \mathsf{N}(\boldsymbol{0}, \boldsymbol{\Sigma})$, $\Sigma_{jj'} = \exp(-|j - j'|)$ (dependent). The response vector $\boldsymbol{y}$ is generated from $\boldsymbol{y} \sim \mathsf{N}(\boldsymbol{X}\boldsymbol{\beta}, \mathbf{I}_n)$, with $\boldsymbol{\beta}$ having only 10 nonzero entries at the first 10 coordinates: $\boldsymbol{\beta} = \text{SNR}\sqrt{(\log p)/n}(2, -3, 2, 2, -3, 3, -2, 3, -2, 3)$ where $\text{SNR} \in \{2, 4\}$ is a signal-to-noise ratio. For each of the four settings, we simulate 50 replicate datasets. Per each replicate, algorithms are randomly initialized with state $\gamma_0$ such that $\gamma_0 \cap \gamma^* = \emptyset$ and $d_{\mathrm{H}}(\gamma_0, \gamma^*) = 20$ which implies $H = 20$ is the minimum required hitting iteration. All hyperparameter setups, the number of MCMC iterations, and other details are deferred to Appendix B.1.

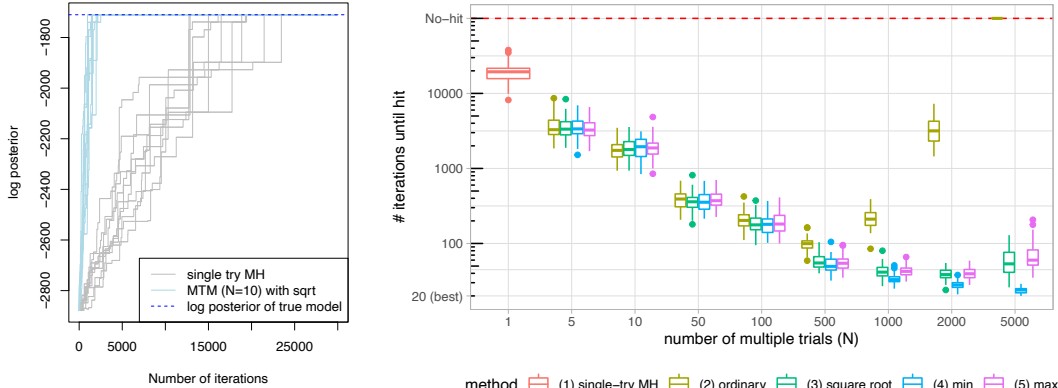

Figure 3: BVS simulation results for independent design and SNR = 4. (Left) Trace plots of unnormalized log-posterior probability using single-try MH and MTM ($N = 10$) with square root weighting function. (Right) Boxplot of $H$, the number of iterations until hit, against different choices of weight function and $N$ based on 50 replicates.

Results under independent design with SNR = 4 are summarized in Figure 3. The trace plot shows that single-try MH reaches the true state $\gamma^*$ at around 20,000 iterations, whereas the MTM with $N = 10$ reaches the true state $\gamma^*$ at around 2,000 iterations, smaller by a factor of 10. The boxplot confirms our findings that a larger number of trials $N$ is not always desirable, especially when using the ordinary weight function $w_{\mathrm{ord}}$. The proposed locally balanced weight functions using three different choices of $h$ perform much better than $w_{\mathrm{ord}}$ for large $N$. Especially, the performance of $w_{\min}$ seems to be robust to the choice of large $N$ in the high SNR case. This can be explained by the fact that the denominator in (2) is often larger in high SNR data, and the choice of $\min\{1, \pi(y)/\pi(x)\}$ limits its growth; see also Section 3.1. The choice of $N$ suggested from Algorithm 2 with $\psi = 0.9$ is overall reasonable, which has a median of 349 over 50 datasets. The detailed results for other settings are available in Appendix B.1.

## 4.2 Stochastic block model (SBM)

SBM [20] is a popular generative model of an undirected graph which assumes a block structure of edge connection probabilities $\boldsymbol{Q}$ to describe community structure. We consider Bayesian SBM [34] by assigning prior on $\boldsymbol{Q}$ and partition $\mathbf{z} = (z_1, \ldots, z_p) \in \{1, \ldots, K\}^p$ with fixed number of blocks $K$, where $z_i$ indicates membership label of $i$-th node. The main goal is to find partition $\mathbf{z}$ (up to a label permutation) that best describes the graph, denoted as an adjacency matrix $\boldsymbol{A} \in \{0, 1\}^{p \times p}$. We follow the prior of [55]:

$$\text{Edge appearance:} \quad A_{ij} \mid \boldsymbol{Q}, \mathbf{z} \overset{\text{ind}}{\sim} \mathsf{Bernoulli}(Q_{z_i z_j}), 1 \le i < j \le p$$

$$\text{Prior on blockwise probabilities:} \quad Q_{uv} \overset{\text{iid}}{\sim} \mathsf{Beta}(\kappa_1, \kappa_2), 1 \le u \le v \le K$$

$$\text{Prior on partition:} \quad \pi(\mathbf{z}) \propto \mathbb{1}(\mathbf{z} \in S_\alpha),$$

with restricted partition space $S_\alpha = \{\mathbf{z} : \sum_{i=1}^p \mathbb{1}(z_i = u) \in \left[\frac{p}{\alpha K}, \frac{\alpha p}{K}\right] \text{ for all } u = 1, \ldots, K\}$ for some $\alpha > 0$, which excludes partitions whose block sizes differ too much. We sample $\mathbf{z}$ from the posterior $\pi(\mathbf{z} \mid \boldsymbol{A})$ where $\boldsymbol{Q}$ is marginalized out; see Appendix B.2 for details. We consider the following proposal where $\tilde{d}_{\mathrm{H}}$ is a permutation-invariant Hamming distance [51]:

$$\mathbf{K}_{\mathrm{RW}}(\mathbf{z}, \mathbf{z}') = 1/(p(K-1))\mathbb{1}_{\mathcal{N}(\mathbf{z})}(\mathbf{z}'), \quad \mathcal{N}(\mathbf{z}) = \{\mathbf{z}' : \tilde{d}_{\mathrm{H}}(\mathbf{z}, \mathbf{z}') = 1\} \quad \text{(single flip).}$$

With the number of nodes $p = 1000$, we consider two different settings of the blocks $K \in \{2, 5\}$ with true partition $\mathbf{z}^* = (1, \ldots, 1, \cdots, K, \ldots, K)$ being balanced ($p/K$ times each) which ensures $\mathbf{z}^* \in S_\alpha$. In each setting of $K$, we generate a graph from the homogeneous SBM [16, 55], where $Q_{uv} = a$ if $u = v$ and $Q_{uv} = b$ otherwise. We choose probability pairs $(a, b)$ based on the Chernoff-Hellinger divergence (CH) [1], defined as $\mathrm{CH} := p(\sqrt{a} - \sqrt{b})^2/(K \log p)$ for $a, b \asymp (\log p)/p$, which can be interpreted as a signal-to-noise ratio. Our choices are $\mathrm{CH} \approx 2$ (weak signal) and $\mathrm{CH} \approx 10$ (high signal), both being greater than 1 which is necessary to satisfy posterior consistency [1, 51, 55]. For each of the the four combinations of $(K, \mathrm{CH})$, we simulate 50 replicate datasets. For each replicate, algorithms are randomly initialized with state $\mathbf{z}_0$ such that $\tilde{d}_{\mathrm{H}}(\mathbf{z}_0, \mathbf{z}^*) = 400$ which implies $H = 400$ is the minimum required hitting iteration. All hyperparameter setups, the number of MCMC iterations, and other details are deferred to Appendix B.2.

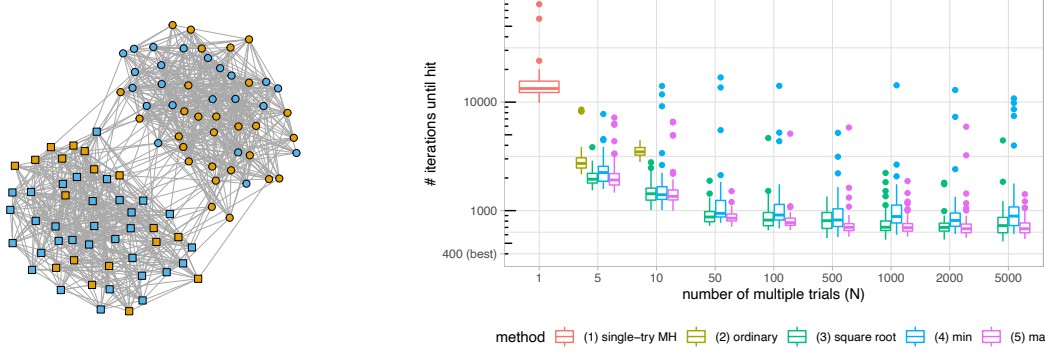

Figure 4: (Left) An example graph generated from SBM with $p = 100$, $K = 2$, $a = 0.32$, and $b = 0.02$ so that $\mathrm{CH} \approx 2$. Node shape (square, circle) denotes the true partition. Node color (blue, yellow) represents the initial state $\mathbf{z}_0$ satisfying $\tilde{d}_{\mathrm{H}}(\mathbf{z}_0, \mathbf{z}^*) = 40$. (Right) Boxplot of $H$ against different weight functions and $N$ based on 50 replicates when $p = 1000$, $K = 2$, and $\mathrm{CH} \approx 2$.

Results for $K = 2$ and $\mathrm{CH} \approx 2$ are summarized in Figure 4. The boxplot of $H$ is similar to that of Figure 3: $w_{\mathrm{ord}}$ quickly deteriorates as $N$ increases, but the locally balanced weight functions mostly converge with a larger choice of $N$, although some Markov chains fail to converge due to the low CH. Especially, now the performance of $w_{\max}$ or $w_{\mathrm{sqrt}}$ is slightly better than $w_{\min}$ when $N$ is large. This can be explained by the fact that in low signal case ($\mathrm{CH} \approx 2$), there are few neighboring states that have high probability ratios. It is important to catch such states to increase the acceptance ratio (2), and the $w_{\min}$ cannot do this efficiently as it is upper bounded by 1. The choice of $N$ suggested from Algorithm 2 with $\psi = 0.9$ has a median of 15 over 50 datasets. The detailed simulation results for other settings are available at Appendix B.2.

Table 1: Summary of $T_H$, median wall-clock time (in seconds) until the Markov chain hit the highest posterior state, for the previous two examples using the square root weight function over 50 replicates.

| $N$ | | 1 | 5 | 10 | 50 | 100 | 500 | 1000 | 2000 | 5000 |
|---|---|---|---|---|---|---|---|---|---|---|
| $T_H$ | BVS, indep, SNR$= 4$ | 1.30 | 0.81 | 0.46 | 0.11 | 0.07 | **0.04** | 0.05 | 0.09 | 0.30 |
| | SBM, $K = 2$, CH $\approx 2$ | 0.82 | 0.40 | **0.35** | 0.48 | 0.75 | 1.49 | 2.15 | 3.78 | 9.30 |

Finally, Table 1 provides the summary of wall-clock hitting time $T_H$ for the previous two examples, which shows the clear computational benefit of MTM over the single-try MH thanks to the parallelism. As $N$ increases, computation becomes more demanding compared to the decreasing rate of $H$, so the optimal choice of $N$ with respect to $T_H$ is often less than the minimizer of $H$.

**Spatial clustering models and additional information.** In Appendix B.3, we present simulation study results with spatial clustering models where the key findings align with those from BVS and SBM. In Appendix B.4, we also analyze the behavior of MTM algorithms on multimodal target distributions. The code is available at https://github.com/changwoo-lee/rapidMTM.

## 5 Real data applications and discussion

**Real data applications.** We carry out two real data application analyses to corroborate our findings beyond the scope of the simulation settings. The performance measures suggested in Section 4, the hitting iteration $H$ and the corresponding wall-clock hitting time $T_H$, are no longer available for real data analysis since we cannot identify the true data generating state $x^*$. Instead, we evaluate the performance based on the acceptance rate and the number of unique states visited. Both real data analyses confirm our theory: the performance deteriorates significantly as the number of trials $N$ grows when using $w_{\text{ord}}$, but it does not when using $w_{\text{sqrt}}$, $w_{\text{min}}$, or $w_{\text{max}}$.

The first real data application is to find a subset of genetic variants from a genome-wide association study (GWAS) dataset that best explains the cup-to-disk ratio averaged over two eyes, which is used to assess the risk of glaucoma. We employ BVS model described in Section 4.1, with the number of samples $n = 5418$ and the number of gene variants $p = 7255$ which are selected after the preliminary screening procedure described in [54, Section 6]. We tabulate the acceptance rate and the number of unique states visited along the different number of trials $N$, and also report the posterior inclusion probabilities of the top 10 genetic variants in Appendix C.1 and E with discussion.

The other application is to learn the underlying directed acyclic graph (DAG) model for the single-cell RNA dataset on Alzheimer's disease [22]. We preprocess the dataset as [10, Section 6], which yields the sample size $n = 1666$ and 73 genes, and we utilize the DAG model described in [10, Section 2.2], whose model size is equal to $73! \approx 4.5 \times 10^{105}$. We defer the performance result and other details (including the log-posterior trace plots for 4 different weight functions) to Appendix C.2.

**Discussion and future work.** We prove that, under some assumptions, the mixing time bound of the MTM algorithm is smaller than that of the MH algorithm by a factor of the number of trials. Motivated by the observation that popular choices of weight function can cause mixing problems, we propose locally balanced weight functions for which a mixing time bound is proven. We also suggest a theoretically guided choice of $N$ for practical use. Future research may investigate the mixing time of other generalizations of MTM such as the case with correlated weight functions. It will also be of interest to extend MTM and analyze its mixing time to better handle the potential multimodality in the target distribution, by combining it with techniques such as annealing or tempering [9].

## Acknowledgments and Disclosure of Funding

We thank James Cai for providing us with the processed single-cell data set on Alzheimer's disease. We thank the submitters and participants of the two dbGaP studies (phs000308.v1.p1 and phs000238.v1.p1), which were funded by NIH. We thank Florian Maire, Giacomo Zanella, and Philippe Gagnon for the helpful discussion at ISBA 2022. The research of Changwoo Lee, Zhao Tang Luo, and Huiyan Sang was partially supported by NSF DMS-1854655 and DMS-2210456.

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
