# Appendices

In Appendix A, we provide proofs of Proposition 1, Proposition 2, and Theorem 1 in the main text. In Appendix B, we provide more details of the Bayesian variable selection (BVS) and stochastic block model (SBM) in Section 4 as well as a detailed simulation study on the spatial clustering model (SCM). In addition, we study the performance of multiple-try Metropolis for the case with multimodal target distributions, following the BVS simulation setting of [54]. In Appendix C, we present details of two real data application analyses. In Appendix D, we add a more detailed discussion on parallelization, state space of interest, and the behavior of MTM on continuous state space. Finally, we provide additional tables on the real data analysis results in Appendix E.

## A   A path method for proving the mixing time bound for multiple-try Metropolis algorithm

### A.1   Proof of Proposition 1

This section aims to provide a summary of the existing results on proving mixing time bound via path methods. We refer readers to [13, 25, 42, 43] for more details. Let $\mathbf{P}(x, y)$ denote the transition probability for an irreducible, aperiodic chain on the finite state space $\mathcal{X}$. Assume $\mathbf{P}$ satisfies the detailed balance condition with respect to the probability distribution $\pi$, that is, $\pi(x)\mathbf{P}(x, y) = \pi(y)\mathbf{P}(y, x)$ for $x, y \in \mathcal{X}$, which leads to $\pi$ being stationary for $\mathbf{P}$ [25, Proposition 1.20]. $\mathbf{P}$ may be thought as a $|\mathcal{X}| \times |\mathcal{X}|$ stochastic matrix, which means $\mathbf{P}(x, y) \geq 0$ and $\sum_{z \in \mathcal{X}} \mathbf{P}(x, z) = 1$ for all $x, y \in \mathcal{X}$, and $\pi$ can be regarded as a $|\mathcal{X}|$-dimensional stochastic vector since $\sum_{x \in \mathcal{X}} \pi(x) = 1$. By the spectral decomposition, we can sort the eigenvalues of $\mathbf{P}$ as

$$1 = \lambda_0 > \lambda_1 \geq \cdots \geq \lambda_{|\mathcal{X}|-1} > -1,$$

due to $\mathbf{P}$ being stochastic, irreducible and aperiodic [25, Lemma 12.1]. Let $\lambda_{\max} = \max\{\lambda_1, |\lambda_{|\mathcal{X}|-1}|\}$. We say $\mathrm{Gap}(\mathbf{P}) = 1 - \lambda_{\max}$ is the *spectral gap* of the chain $\mathbf{P}$. Intuitively, if the spectral gap is close to zero, the chain requires a large number of steps to be close to the stationary distribution in total variation distance. The following lemma draws the connection between the spectral gap and mixing time defined in Section 2.3.

**Lemma 1.** *Let $t_{\mathrm{mix}}(\epsilon)$ denote $\epsilon$-mixing time defined in Section 2.3, then*

$$t_{\mathrm{mix}}(\epsilon) \leq 2\{\mathrm{Gap}(\mathbf{P})\}^{-1}\left[\log(1/\epsilon) + \log\left\{\min_{x \in \mathcal{X}}\pi(x)\right\}^{-1}\right].$$

*Proof.* We consider $\mathbf{P}_{\mathrm{lazy}} = (\mathbf{I} + \mathbf{P})/2$ so that all eigenvalues of $\mathbf{P}_{\mathrm{lazy}}$ are positive and the spectral gap becomes $\mathrm{Gap}(\mathbf{P}_{\mathrm{lazy}}) = 1 - \lambda_1$. By (1.10) of [13, Proposition 3],

$$4\max_{x \in \mathcal{X}}\|\mathbf{P}_{\mathrm{lazy}}^t(x, \cdot) - \pi(\cdot)\|_{\mathrm{TV}}^2 \leq \max_{x \in \mathcal{X}}\{(1 - \pi(x))/\pi(x)\}\exp[-2t\mathrm{Gap}(\mathbf{P}_{\mathrm{lazy}})],$$

for $t \in \mathbb{N}$. We set $\max_{x \in \mathcal{X}}\{(1 - \pi(x))/\pi(x)\}\exp[-2t\mathrm{Gap}(\mathbf{P}_{\mathrm{lazy}})] \leq 4\epsilon^2$, and solving the inequality with $t$ gives

$$t \geq \{\mathrm{Gap}(\mathbf{P}_{\mathrm{lazy}})\}^{-1}\left(\frac{1}{2}\max_{x \in \mathcal{X}}\log(\pi(x)^{-1} - 1) - \log 2 + \log(1/\epsilon)\right).$$

Since

$$\{\mathrm{Gap}(\mathbf{P}_{\mathrm{lazy}})\}^{-1}\left(\frac{1}{2}\max_{x \in \mathcal{X}}\log(\pi(x)^{-1} - 1) - \log 2 + \log(1/\epsilon)\right)$$

$$\leq \{\mathrm{Gap}(\mathbf{P}_{\mathrm{lazy}})\}^{-1}\left[\log(1/\epsilon) + \log\left\{\min_{x \in \mathcal{X}}\pi(x)\right\}^{-1}\right]$$

$$\leq 2\{\mathrm{Gap}(\mathbf{P})\}^{-1}\left[\log(1/\epsilon) + \log\left\{\min_{x \in \mathcal{X}}\pi(x)\right\}^{-1}\right],$$

it follows that to achieve "$\epsilon$-mixing", it suffices to choose

$$t \geq 2\{\text{Gap}(\mathbf{P})\}^{-1} \left[ \log(1/\epsilon) + \log \left\{ \min_{x \in \mathcal{X}} \pi(x) \right\}^{-1} \right].$$

This concludes the proof. $\qquad\square$

Our next interest is to find the lower bound of the spectral gap, which leads to the upper bound of the mixing time. It is often the case that the functional analysis tool is useful to establish such bound. Let $\mathbb{R}^{\mathcal{X}} = \{f\colon \mathcal{X} \to \mathbb{R}\}$ and $\ell^2(\pi) \subset \mathbb{R}^{\mathcal{X}}$ be the vector space equipped with an inner product $\langle \cdot, \cdot \rangle_\pi$, which is defined by $\langle f_1, f_2 \rangle_\pi = \sum_{x \in \mathcal{X}} f_1(x) f_2(x) \pi(x)$ for all $f_1, f_2 \in \ell^2(\pi)$. We can regard the transition probability $\mathbf{P}$ as a function operator in $\ell^2(\pi)$, which can be defined as $\mathbf{P}f(x) = \sum_{y \in \mathcal{X}} \mathbf{P}(x, y) f(y)$. We define the *Dirichlet form* associated to the pair $(\mathbf{P}, \pi)$ by

$$\mathcal{E}(f_1, f_2) = \langle (\mathbf{I} - \mathbf{P}) f_1, f_2 \rangle_\pi \quad \text{for } f_1, f_2 \in \ell^2(\pi).$$

By the reversibility of $\mathbf{P}$, we can easily check that $\mathcal{E}(f) = \mathcal{E}(f, f) = \frac{1}{2} \sum_{x,y \in \mathcal{X}} [f(x) - f(y)]^2 \pi(x) \mathbf{P}(x, y)$. The spectral gap can be defined using the Dirichlet form as follows [25, Remark 13.8]:

$$\text{Gap}(\mathbf{P}) = \min_{\substack{f \in \ell^2(\pi) \\ \text{Var}_\pi(f) \neq 0}} \frac{\mathcal{E}(f)}{\text{Var}_\pi(f)},$$

where $\text{Var}_\pi(f) = \sum_{x \in \mathcal{X}} (f(x) - \mathbb{E}_\pi f)^2 \pi(x)$. This definition also has a link to the famous *Poincaré inequality* [4], that is, $\text{Var}_\pi(f) \leq C \mathcal{E}(f)$ for all $f \in \ell^2(\pi)$, because the smallest constant $C$ is equal to $\{\text{Gap}(\mathbf{P})\}^{-1}$. The next lemma uses Poincaré inequality and an arbitrary path ensemble $\Delta$ defined in the main text.

**Lemma 2** (Corollory 6, [43]). *For an arbitrary path ensemble $\Delta$,*

$$\text{Gap}(\mathbf{P}) \geq \frac{1}{\rho(\Delta) l(\Delta)}.$$

*where $\ell(\Delta) = \max_{x,y} |\delta(x, y)|$ and*

$$\rho(\Delta) = \max_{(u,v) \in E} \frac{1}{\pi(u) \mathbf{P}(u, v)} \sum_{x,y : \delta(x,y) \ni (u,v)} \pi(x) \pi(y)$$

*Proof.* We follow the proof given in [42, Theorem 3.2.1]. For each $(x, y) \in \mathcal{X} \times \mathcal{X}$ and for any function $f \in \mathbb{R}^{\mathcal{X}}$, we can write $f(y) - f(x) = \sum_{(u,v) \in \delta(x,y)} f(v) - f(u)$. By using Cauchy-Schwarz, multiplying $\pi(x)\pi(y)/2$, and summing over $x$ and $y$,

$$|f(y) - f(x)|^2 \leq |\delta(x, y)| \sum_{(u,v) \in \delta(x,y)} |f(v) - f(u)|^2$$

$$\implies \underbrace{\frac{1}{2} \sum_{x,y} |f(y) - f(x)|^2 \pi(x) \pi(y)}_{=\text{Var}_\pi(f)} \leq \frac{1}{2} \sum_{x,y} |\delta(x, y)| \sum_{(u,v) \in \delta(x,y)} |f(v) - f(u)|^2 \pi(x) \pi(y),$$

where the right-hand side becomes

$$\frac{1}{2} \sum_{(u,v) \in E} \left\{ \frac{1}{\pi(u) \mathbf{P}(u, v)} \sum_{x,y : \delta(x,y) \ni (u,v)} |\delta(x, y)| \pi(x) \pi(y) \right\} |f(v) - f(u)|^2 \pi(u) \mathbf{P}(u, v)$$

$$\leq \underbrace{\max_{(u,v) \in E} \left\{ \frac{1}{\pi(u) \mathbf{P}(u, v)} \sum_{x,y : \delta(x,y) \ni (u,v)} \pi(x) \pi(y) \right\}}_{\rho(\Delta)} \underbrace{\left( \max_{x,y} |\delta(x, y)| \right)}_{\ell(\Delta)} \underbrace{\left( \frac{1}{2} \sum_{(u,v) \in E} |f(v) - f(u)|^2 \pi(u) \mathbf{P}(u, v) \right)}_{\mathcal{E}(f)}.$$

This satisfies the Poincaré inequality, which yields the conclusion. $\qquad\square$

By combining the results of Lemma 1 and Lemma 2, we get the conclusion of Proposition 1.

## A.2 Proof of Proposition 2

*Proof.* Recall that the form of the weight function suggested in [27] is given by

$$w(y \mid x) = \pi(y)\mathbf{K}_{\text{RW}}(y, x)\lambda(y, x),$$

where $\lambda(x, y) = \lambda(y, x)$ is a non-negative symmetric function in $x$ and $y$, and satisfies $\lambda(x, y) > 0$ whenever $\mathbf{K}_{\text{RW}}(x, y) > 0$. If we put

$$\lambda(x, y) = \frac{1}{\pi(y)\mathbf{K}_{\text{RW}}(y, x)} h\left(\frac{\pi(y)\mathbf{K}_{\text{RW}}(y, x)}{\pi(x)\mathbf{K}_{\text{RW}}(x, y)}\right),$$

it is easy to check that the conditions are met. $\qquad\square$

## A.3 Proof of Theorem 1

In this section, we prove our main result by using Proposition 1. The main step of the proof is identifying the path ensemble $\Delta^*$ that makes the mixing time bound tight. To this end, we need to choose exactly one path $\delta^*(x, y)$ for each tuple $(x, y) \in \mathcal{X} \times \mathcal{X}$. (Note that $x$ and $y$ cannot be identical by the definition of path). From the intuition described in Section 2.3, an edge $(u, v)$ has a large capacity if $\pi(u)$ and $\pi(v)$ are large. For example, if an edge contains the highest posterior state $x^*$, we can let the edge be traversed by a large number of paths. Given an edge with a small capacity, however, we need to ensure that the edge overlies with a small number of paths. Importantly, we do not let a path $\delta^*(x, y)$ pass through an edge with a small capacity if both $\pi(x)$ and $\pi(y)$ are large, so that the edge can maintain a small unit flow size. We may envision the topography of the path ensemble; $x^*$ becomes the hub, while states with low posterior probability are located on the outskirt.

We construct the path ensemble $\Delta^*$ according to the description above. The construction of $\Delta^*$ is similar to that of [47] and [52]. With a neighborhood relation $\mathcal{N}$ that satisfies the conditions in Theorem 1, $g \colon \mathcal{X} \to \mathcal{X}$,

$$g(x) = \begin{cases} \arg\max_{x' \in \mathcal{N}(x)} \pi(x')^1 & \text{if } x \neq x^*, \\ x^* & \text{otherwise.} \end{cases} \tag{7}$$

By Condition (i) in Theorem 1, there exists $m \in \mathbb{N}$ such that $g^m(x) = (\overbrace{g \circ \cdots \circ g}^{m \text{ times}})(x) = x^*$ for any $x \in \mathcal{X}$ and $x^*$ is the only fixed point of $g$. ($x^*$ can be thought as an attractor in dynamic systems.) For all $x, y \in \mathcal{X}$ with $x \neq y$, we have three cases; (i) $g^m(x) = y$ for some $m \in \mathbb{N} \setminus \{0\}$, (ii) $g^m(y) = x$ for some $m \in \mathbb{N} \setminus \{0\}$, or (iii) neither (i) nor (ii). If $(x, y) \in \mathcal{X} \times \mathcal{X}$ belongs to (i), we define $\delta^*(x, y) = (x, g(x), \ldots, g^m(x) = y)$. Similarly, if $(x, y) \in \mathcal{X} \times \mathcal{X}$ belongs to (ii), let $\delta^*(x, y) = (x = g^m(y), \ldots, g(y), y)$. For the case (iii), if $m_1, m_2 \in \mathbb{N} \setminus \{0\}$ are the minimum numbers that satisfy $g^{m_1}(x) = x^*, g^{m_2}(y) = x^*$, respectively, we let $\delta^*(x, y) = (x, g(x), \ldots, g^{m_1}(x) = x^* = g^{m_2}(y), \ldots, g(y), y)$. This yields the path ensemble $\Delta^*$. We provide a toy example on how to construct $g$ and any path $\delta^*(x, y)$ for each tuple $(x, y) \in \mathcal{X} \times \mathcal{X}$ associated with $g$ in Appendix A.5.

Next, we make a bound for the congestion parameter $\rho(\Delta^*)$. We let $\Lambda(u) = \{x \in \mathcal{X} : u = g^k(x), k \in \mathbb{N}\}$ denote the ancestor set of $u$ with respect to $g$. If $(u, v) \in \delta^*(x, y)$ for some $x, y \in \mathcal{X}$, we can easily verify that $x \in \Lambda(u)$ by the construction of $\delta^*$. This implies $\{(x, y) \in \mathcal{X} \times \mathcal{X} : (u, v) \in \delta^*(x, y)\} \subseteq \Lambda(u) \times \mathcal{X}$. It follows that

$$\rho(\Delta^*) \leq \max_{(u,v):v=g(u)} \frac{1}{\pi(u)\mathbf{P}(u, v)} \sum_{(x,y) \in \Lambda(u) \times \mathcal{X}} \pi(x)\pi(y)$$

$$= \max_{(u,v):v=g(u)} \frac{1}{\pi(u)\mathbf{P}(u, v)} \left(\sum_{x \in \Lambda(u)} \pi(x)\right)\left(\sum_{y \in \mathcal{X}} \pi(y)\right)$$

$$= \max_{(u,v):v=g(u)} \frac{\pi(\Lambda(u))}{\pi(u)\mathbf{P}(u, v)}$$

$$\leq \max_{(u,v):v=g(u)} \frac{1}{(1 - p^{-(t_2 - t_4)})\mathbf{P}(u, v)}.$$

---

[1]If multiple states tie, we randomly pick one of them.

The last inequality holds by the following. Let $g^{-k}(u) = \{x \in \mathcal{X} : g^k(x) = u, g^{k-1}(x) \neq u\}$ for $k \in \mathbb{N}$, then $\Lambda(u) = \uplus_{k \in \mathbb{N}} g^{-k}(u)$. By Condition (ii) in Theorem 1, we have $|g^{-k}(u)| \leq p^{kt_4}$ which yields,

$$\frac{\pi(\Lambda(u))}{\pi(u)} = \sum_{k \in \mathbb{N}} \frac{\pi(g^{-k}(u))}{\pi(u)} \leq \sum_{k \in \mathbb{N}} p^{-k(t_2 - t_4)} = \frac{1}{1 - p^{-(t_2 - t_4)}},$$

where we use Condition (i) and the definition of $g$, and $t_2 > t_4$ in Theorem 1.

Finally, we show that $\mathbf{P}(x, g(x)) \geq C' \frac{N}{p^{t_4}}$ for $x \neq x^*$ and for some universal constant $C' > 0$. Recall that (see also [27, Theorem 1])

$$\mathbf{P}(x, g(x)) = N \sum_{y_1, \ldots, y_{N-1}} \sum_{x_1^\star, \ldots, x_{N-1}^\star} \frac{w(g(x) \mid x)}{w(g(x) \mid x) + \sum_{j=1}^{N-1} w(y_j \mid x)} \min \left\{ 1, \frac{w(g(x) \mid x) + \sum_{j=1}^{N-1} w(y_j \mid x)}{w(x \mid g(x)) + \sum_{j=1}^{N-1} w(x_j^\star \mid g(x))} \right\} \times$$

$$\mathbf{K}_{\mathrm{RW}}(x, g(x)) \mathbf{K}_{\mathrm{RW}}(x, y_1) \cdots \mathbf{K}_{\mathrm{RW}}(x, y_{N-1}) \mathbf{K}_{\mathrm{RW}}(g(x), x_1^\star) \cdots \mathbf{K}_{\mathrm{RW}}(g(x), x_{N-1}^\star)$$

$$\geq \left( N \sum_{y_1, \ldots, y_{N-1}} \frac{w(g(x) \mid x)}{w(g(x) \mid x) + \sum_{j=1}^{N-1} w(y_j \mid x)} \mathbf{K}_{\mathrm{RW}}(x, g(x)) \mathbf{K}_{\mathrm{RW}}(x, y_1) \cdots \mathbf{K}_{\mathrm{RW}}(x, y_{N-1}) \right) \times$$

$$\left( \sum_{x_1^\star, \ldots, x_{N-1}^\star} \min \left\{ 1, \frac{w(g(x) \mid x)}{w(x \mid g(x)) + \sum_{j=1}^{N-1} w(x_j^\star \mid g(x))} \right\} \mathbf{K}_{\mathrm{RW}}(g(x), x_1^\star) \cdots \mathbf{K}_{\mathrm{RW}}(g(x), x_{N-1}^\star) \right),$$

and we denote the first and the second terms of the right-hand side by

$$\mathbf{K}(x, g(x)) = N \sum_{y_1, \ldots, y_{N-1}} \frac{w(g(x) \mid x)}{w(g(x) \mid x) + \sum_{j=1}^{N-1} w(y_j \mid x)} \mathbf{K}_{\mathrm{RW}}(x, g(x)) \mathbf{K}_{\mathrm{RW}}(x, y_1) \cdots \mathbf{K}_{\mathrm{RW}}(x, y_{N-1}),$$

$$(8)$$

$$\eta(x, g(x)) = \sum_{x_1^\star, \ldots, x_{N-1}^\star} \min \left\{ 1, \frac{w(g(x) \mid x)}{w(x \mid g(x)) + \sum_{j=1}^{N-1} w(x_j^\star \mid g(x))} \right\} \mathbf{K}_{\mathrm{RW}}(g(x), x_1^\star) \cdots \mathbf{K}_{\mathrm{RW}}(g(x), x_{N-1}^\star).$$

$$(9)$$

Hence, we have $\mathbf{P}(x, g(x)) \geq \mathbf{K}(x, g(x)) \eta(x, g(x))$. We remark that the formulation of $\mathbf{K}(x, g(x))$ in (8) is based on the exchangeability, where the $N$-th trial state $y_N = g(x)$ is assumed to be selected as a proposal state and its probability is multiplied by $N$. However, we will not utilize such exchangeability to calculate a lower bound of $\mathbf{K}(x, g(x))$. Instead, we define an event $A$ that selects $g(x)$ as the proposal from Step 1 and Step 2 in Algorithm 1, so that $\mathbb{P}(A) = \mathbf{K}(x, g(x))$. We aim to lower bound the $\mathbb{P}(A)$ by using the law of total probability. To this end, we introduce the event $F$ for Step 1 in Algorithm 1 that we include the state $g(x)$ at least once among the $N$ trials while we don't sample any "high" posterior states of the neighborhood of $x$ for the rest of trials, i.e. they do not belong to the set $\mathcal{S}(x)$. Using conditional probability rule, the probability of the event $F$ is equal to

$$\mathbb{P}(F) = \left( \frac{|\mathcal{N}(x)| - |\mathcal{S}(x)| + 1}{|\mathcal{N}(x)|} \right)^N \left( 1 - \left( \frac{|\mathcal{N}(x)| - |\mathcal{S}(x)|}{|\mathcal{N}(x)| - |\mathcal{S}(x)| + 1} \right)^N \right), \quad (10)$$

where we take the probability over the uniform samples in Step 1 in Algorithm 1. Using the inequality $a^n - b^n = (a - b)(a^{n-1} + a^{n-2}b + \cdots + b^{n-1}) \geq (a - b)nb^{n-1}$ for any $a \geq b \geq 0$, we find that the lower bound of $\mathbb{P}(F)$ can be obtained by

$$\mathbb{P}(F) = \left( \left( 1 - \frac{|\mathcal{S}(x)| - 1}{|\mathcal{N}(x)|} \right)^N - \left( 1 - \frac{|\mathcal{S}(x)|}{|\mathcal{N}(x)|} \right)^N \right)$$

$$\geq \frac{N}{|\mathcal{N}(x)|} \left( 1 - \frac{|\mathcal{S}(x)|}{|\mathcal{N}(x)|} \right)^{N-1}$$

$$\geq \frac{N}{p^{t_4}} \left( 1 - (N - 1) \frac{s_0}{p^{t_3}} \right)$$

$$\geq \frac{N}{p^{t_4}} (1 + o(1)),$$

where Condition (ii) and Bernoulli's inequality are used in the second inequality, and Condition (iii) is used in the last inequality. We further define $F_k$ as the event $F$ with $k$ number of $g(x)$ among $N$ trials. (Note that since we sample trials with replacement in Step 1 of Algorithm 1, we may sample $g(x)$ multiple times.) Observe that $F = \uplus_{k=1}^{N} F_k$. Given the event $F_k$, the probability to select $g(x)$ in (2) of Algorithm 1 is upper bounded by

$$
\begin{aligned}
\mathbb{P}(A \mid F_k) &= \frac{kw(g(x) \mid x)}{kw(g(x) \mid x) + \sum_{i=1}^{N-k} w(y_j \mid x)} \\
&= \left\{ 1 + k^{-1} \sum_{i=1}^{N-k} \frac{w(y_j \mid x)}{w(g(x) \mid x)} \right\}^{-1} \\
&\overset{(\star)}{\geq} \left\{ 1 + N \frac{h(p^{t_1+t_4-t_3})}{h(p^{t_2+t_3-t_4})} \right\}^{-1} = 1 + o(1).
\end{aligned}
$$

To see $(\star)$, we have used the fact that $h$ is a non-decreasing function, Condition (iii), and we have

$$
\frac{\pi(y_j)}{\pi(x)} \cdot \frac{\mathbf{K}_{\mathrm{RW}}(y_j, x)}{\mathbf{K}_{\mathrm{RW}}(x, y_j)} \leq p^{t_1} p^{t_4-t_3},
$$

$$
\frac{\pi(g(x))}{\pi(x)} \cdot \frac{\mathbf{K}_{\mathrm{RW}}(g(x), x)}{\mathbf{K}_{\mathrm{RW}}(x, g(x))} \geq p^{t_2} p^{t_3-t_4},
$$

from Condition (i) and (ii). Note that the right-hand side of the inequality $(\star)$ does not depend on $k$. Using the law of total probability, we yield the lower bound of $\mathbb{P}(A) = \mathbf{K}(x, g(x))$ by combining the previous results:

$$
\begin{aligned}
\mathbf{K}(x, g(x)) &= \mathbb{P}(A|F)\mathbb{P}(F) + \mathbb{P}(A|F^c)\mathbb{P}(F^c) \geq \mathbb{P}(A|F)\mathbb{P}(F) \\
&= \sum_{k=1}^{N} \mathbb{P}(A|F_k)\mathbb{P}(F_k) \geq \sum_{k=1}^{N}(1 + o(1))\mathbb{P}(F_k) \\
&= (1 + o(1))\mathbb{P}(F) \geq \frac{N}{p^{t_4}}(1 + o(1)),
\end{aligned} \tag{11}
$$

Similarly, we can calculate the lower bound of $\eta(x, g(x))$. We consider the event $G$ that we don't select any "high" posterior states of the neighborhood of $g(x)$ for $N-1$ trials, that is, any of them are not in the set $\mathcal{S}(g(x))$. A simple calculation yields

$$
\mathbb{P}(G) \geq \left( \frac{|\mathcal{N}(g(x))| - s_0}{|\mathcal{N}(g(x))|} \right)^{N-1}.
$$

Under the event $G$, on the other hand,

$$
\begin{aligned}
\frac{w(g(x) \mid x)}{w(x \mid g(x)) + \sum_{j=1}^{N-1} w(x_j^\star \mid g(x))} &= \left( \frac{w(x \mid g(x))}{w(g(x) \mid x)} + \sum_{j=1}^{N-1} \frac{w(x_j^\star \mid g(x))}{w(g(x) \mid x)} \right)^{-1} \\
&= \left( \frac{\pi(x)\mathbf{K}_{\mathrm{RW}}(x, g(x))}{\pi(g(x))\mathbf{K}_{\mathrm{RW}}(g(x), x)} + \sum_{j=1}^{N-1} \frac{w(x_j^\star \mid g(x))}{w(g(x) \mid x)} \right)^{-1} \\
&\geq \left( p^{-t_2-t_3+t_4} + (N-1)\frac{h(p^{t_1-t_3+t_4})}{h(p^{t_2+t_3-t_4})} \right)^{-1},
\end{aligned}
$$

where the second equality is due to the property of the balancing function $h(u) = uh(1/u)$ and the last inequality follows from a similar argument as before using Conditions (i), (ii) and

$$
\frac{\pi(x_j^\star)}{\pi(g(x))} \cdot \frac{\mathbf{K}_{\mathrm{RW}}(x_j^\star, g(x))}{\mathbf{K}_{\mathrm{RW}}(g(x), x_j^\star)} \leq p^{t_1} p^{t_4-t_3},
$$

by non-decreasing $h$. Then for $x \neq x^*$, by Conditions (ii), (iii) and we have

$$
\begin{aligned}
\eta(x, g(x)) &\geq \min \left\{ 1, \left( p^{-t_2-t_3+t_4} + (N-1) \frac{h(p^{t_1-t_3+t_4})}{h(p^{t_2+t_3-t_4})} \right)^{-1} \right\} \left( \frac{|\mathcal{N}(g(x))| - s_0}{|\mathcal{N}(g(x))|} \right)^{N-1} \\
&\geq (1 + o(1))(1 - s_0/p^{t_3})^{N-1} \\
&\geq 1 - \frac{N s_0}{p^{t_3}} + o(1) \\
&= 1 + o(1). \tag{12}
\end{aligned}
$$

Combining the lower bounds (11) and (12) leads to $\mathbf{P}(x, g(x)) \geq C' \frac{N}{p^{t_4}}$, which concludes the proof of the theorem.

### A.4 An example on a different weight function.

The example below shows an undesirable behavior of the MTM algorithm if we use a weight function which is not in the class of (4). Here an undesirable behavior means that the acceptance probability is close to zero even when $\pi(x_{\text{prop}}) \gg \pi(x_{\text{curr}})$, where we denote $x_{\text{curr}}, x_{\text{prop}}$ as the current state and the proposed state from (2) in Algorithm 1, respectively.

**Example 1.** We consider the weight function $w(y \mid x) = \pi(y)$. For the sake of simplicity, let $|\mathcal{N}(x)| = p^{t_3}$ and for $x' \in \mathcal{N}(x)$,

$$
\frac{\pi(x')}{\pi(x)} = \begin{cases} p^{t_2}, & \text{for } x' \in \mathcal{S}(x), \\ p^{t_1}/2, & \text{for } x' \notin \mathcal{S}(x), \end{cases}
$$

for all $x \neq x^*$. Let $x_j, x_j^{\star}$ be uniform samples from $\mathcal{N}(x_{\text{curr}})$ and $\mathcal{N}(x_{\text{prop}})$, respectively for $j \in [N-1]$. Assume $x_{\text{prop}} \in \mathcal{S}(x_{\text{curr}})$ to reflect $\pi(x_{\text{prop}}) \gg \pi(x_{\text{curr}})$. Although the conditions on Theorem 1 are met (except for those related to a balancing function $h$), the acceptance probability from $x_{\text{curr}}$ to $x_{\text{prop}}$ is upper bounded as

$$
\begin{aligned}
\alpha(x_{\text{curr}}, x_{\text{prop}}) &= \min \left\{ 1, \frac{\sum_{i=1}^{N-1} \pi(x_i) + \pi(x_{\text{prop}})}{\sum_{i=1}^{N-1} \pi(x_i^{\star}) + \pi(x_{\text{curr}})} \right\} \\
&\leq \frac{N p^{t_2} \pi(x_{\text{curr}})}{(1 + (N-1) p^{t_2+t_1/2}) \pi(x_{\text{curr}})} \\
&\leq \frac{2}{p^{t_1/2}} = o(1).
\end{aligned}
$$

Notice that the acceptance probability $\alpha(x_{\text{curr}}, x_{\text{prop}}) = 1$ in the MH algorithm.

## A.5  A toy example on path construction.

We provide a simple example on how to construct the canonical path ensemble $\Delta^*$ described in A.3. Define $\mathcal{X} = \{0,1\}^3$ as our state space and let $x^* = (1,1,0)$ be the mode of the distribution on $\mathcal{X}$. Let the target distribution $\pi(x) \propto \exp(-d_{\mathrm{H}}(x,x^*))$, where $d_{\mathrm{H}}$ is a Hamming distance. We specify a neighborhood of any $x \in \mathcal{X}$ as $\mathcal{N}(x) = \{y \in \mathcal{X} : d_{\mathrm{H}}(y,x) = 1\}$, where $d_{\mathrm{H}}$ is a Hamming distance. In the left panel of Figure 5, neighboring states are linked to black undirected edges. We follow the rule described in (7) to define a transition function $g : \mathcal{X} \to \mathcal{X}$ and black directed edges indicate the defined moves by a function $g$. The height of each bar indicates $\pi(x)$ associated with the corresponding state $x$. In the right panel of Figure 5, we provide three examples to illustrate the three possible cases to construct a path from the defined transition $g$, described in A.3.

- The red directed edges indicate a path from $(0,0,1)$ to $(1,0,0)$, which corresponds to the case (i), since $(1,0,0) = g^2((0,0,1))$,

- The green directed edge indicates a path from $(1,1,0)$ to $(0,1,0)$, which corresponds to the case (ii), since $(1,1,0) = g((0,1,0))$,

- The blue directed edges indicate a path from $(0,0,1)$ to $(1,1,1)$, which corresponds to the case (iii).

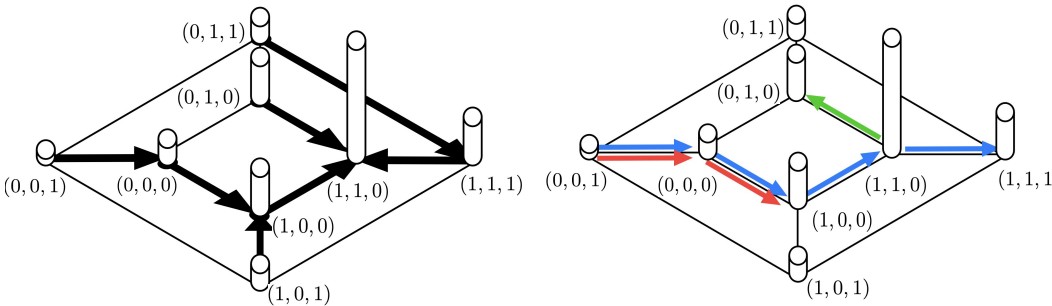

Figure 5: A toy example on path construction. Black undirected edges connect neighboring states and the target distribution $\pi$ is represented as the heights of the cylinders. (Left) Black directed edges indicate the defined moves by a function $g$. (Right) The colored paths exemplify the three possible cases of path construction.

# B  Details of simulation studies

The scope of this paper is to theoretically study the mixing time for the family of MTM algorithms, and hence we mainly focus on experiments to empirically verify our theoretical insights, that the MTM mixing time is smaller by a factor of the number of trials $N$ and that locally balanced weight functions tend to perform better under suitable assumptions. Nevertheless, in some experiments, we compare the MTM algorithm with the locally balanced MH algorithm (denoted as LBMH) [48], as it has been reported to outperform the other state-of-the-art methods. Before describing the details of simulation studies, here we briefly describe the locally balanced MH algorithm.

Specification of LBMH requires balancing function $h$ and uninformed symmetric distribution $\mathbf{K}_{\mathrm{sym}}(x, \cdot)$ supported on $\mathcal{N}(x)$. LBMH chooses a proposal state $y$ from a pointwise informed proposal distribution

$$Q_h(x, y) = (Z_h(x))^{-1} h\left(\pi(y)/\pi(x)\right) \mathbf{K}_{\mathrm{sym}}(x, y), \tag{13}$$

where $Z_h(x) = \sum_{z \in \mathcal{N}(x)} h\left(\pi(z)/\pi(x)\right) \mathbf{K}_{\mathrm{sym}}(x, z)$ is a normalizing constant. Then, $y$ is accepted with probability $\alpha = \min\{1, \frac{\pi(y)Q_h(y,x)}{\pi(x)Q_h(x,y)}\} = \min\{1, \frac{Z_h(x)}{Z_h(y)}\}$, by defintion of balancing function $h(u) = uh(1/u)$ and symmetry of $\mathbf{K}_{\mathrm{sym}}$. Unlike MTM where a subset of $\mathcal{N}(x)$ is selected (with replacement) as a trial and choose a proposal among them, LBMH needs to evaluate $h(\pi(y)/\pi(x))$ for *all* $y \in \mathcal{N}(x)$ to get a proposal state $y$ which can be viewed as an exhaustive search of $\mathcal{N}(x)$. In terms of computation, MTM requires calculating $2N - 1$ weight functions at each iteration where $N$ can be chosen at one's disposal, LBMH requires calculating $|\mathcal{N}(y)|$ number of ratios to calculate $Z_h(y)$ at each iteration, where proposal probabilities $\{Q_h(x, y) : y \in \mathcal{N}(x)\}$ and normalizing constant $Z_h(x)$ can be saved and reused from the previous iteration. Since random walk proposals in BVS and SBM examples are both symmetric, we compare MTM with LBMH by letting $\mathbf{K}_{\mathrm{sym}} = \mathbf{K}_{\mathrm{RW}}$ with three different balancing functions: $h(u) = \sqrt{u}$, $h(u) = \min\{1, u\}$ and $h(u) = \max\{1, u\}$ (corresponding to $w_{\mathrm{sqrt}}, w_{\mathrm{min}}, w_{\mathrm{max}}$ respectively).

## B.1  Details of Bayesian variable selection (BVS)

After marginalizing out $\boldsymbol{\beta}$ and $\phi$, the posterior distribution $\pi(\gamma \,|\, \boldsymbol{y})$ is written as [47, §A.1]

$$\pi(\gamma \,|\, \boldsymbol{y}) = C \cdot \frac{1}{p^{\kappa|\gamma|}(1 + \mathscr{G})^{|\gamma|/2}} \mathrm{SSR}(\gamma)^{-n/2} \mathbb{1}(|\gamma| \le s_{\mathrm{max}}), \tag{14}$$

where $\mathrm{SSR}(\gamma) = \boldsymbol{y}^\top \left(\mathbf{I}_n - \frac{\mathscr{G}}{\mathscr{G}+1} \boldsymbol{X}_\gamma \left(\boldsymbol{X}_\gamma^\top \boldsymbol{X}_\gamma\right)^{-1} \boldsymbol{X}_\gamma^\top\right) \boldsymbol{y}$ is a term having a similar role as a sum of squared residuals and $C$ is a normalizing constant.

**MCMC setup.** Hyperparameters are specified as $\mathscr{G} = p^3 = 5000^3$, $\kappa = 2$, and $s_{\mathrm{max}} = 100$. For each dataset, we run a chain of $10^5$ iteration for single-try MH, $2 \times 10^4$ iteration for MTM with $N = 5$, and $10^4$ iteration for MTM with $N = 5, 10, 50, 100, 500, 1000, 2000, 5000$ using four different weight functions. Algorithms are randomly initialized with state $\gamma_0$ such that $\gamma_0 \cap \gamma^* = \emptyset$ and $d_{\mathrm{H}}(\gamma_0, \gamma^*) = 20$ which implies $H = 20$ is the minimum required hitting iteration. For each simulated dataset, the true data generated model achieves the highest posterior probability ($\gamma^* = x^*$). All simulation studies are performed on a Linux cluster with Intel(R) Xeon(R) Gold 6132 CPU @ 2.60GHz and 96GB memory.

Results from Table 2 show that $H$ decreases roughly by a factor of $N$ until $N = 100$, which confirms our theoretical findings, given that the model setting satisfies that the mixing time is equivalent to the hitting iteration up to constant factors [36]. When $N$ becomes larger, the performance of unscaled weight function $w_{\mathrm{ord}}$ deteriorates and never converges when $N = 5000$. In contrast, locally balanced weight functions generally perform well even when $N$ is large. Table 3 suggests that choosing moderate $N$ is beneficial in terms of computational savings. When the design matrix is correlated and SNR= 2, the result suggests that the chain often stuck when we choose the weight function as $w_{\mathrm{max}}$. Since the shape of the posterior distribution becomes irregular when the SNR is intermediate [47] and design matrix is correlated, to get a more clear insight we further perform additional simulation study when the posterior distribution exhibits multimodality; see Appendix B.4. Finally, under different settings of the design and SNR, the median $N$ estimated from Algorithm 2 using $\psi = 0.9$ over 50 replicate datasets is $\hat{N} = 349$ (indep, SNR=4), 501 (indep, SNR=2), 328 (dep, SNR=4) and 158 (dep, SNR=2).

Table 2: (BVS) Median of $H$, the number of iterations until the chain hit $\gamma^*$ over 50 replicates. Entry with "Fail" indicates that chains never hit $\gamma^*$ in more than half of the replicated datasets.

| | SNR | N | | 1 | 5 | 10 | 50 | 100 | 500 | 1000 | 2000 | 5000 | LBMH |
|---|---|---|---|---|---|---|---|---|---|---|---|---|---|
| ind. | 4 | | $w_{\mathrm{ord}}$ | 19414 | 3283 | 1742 | 392 | 203 | 100 | 211 | 3168 | Fail | N/A |
| | | | $w_{\mathrm{sqrt}}$ | | 3340 | 1787 | 360 | 177 | 55 | 42 | 38 | 54 | Fail |
| | | | $w_{\mathrm{min}}$ | | 3365 | 1948 | 354 | 180 | 50 | 33 | 28 | 24 | 20 |
| | | | $w_{\mathrm{max}}$ | | 3246 | 1876 | 372 | 182 | 54 | 42 | 40 | 60 | Fail |
| | 2 | | $w_{\mathrm{ord}}$ | 20088 | 3684 | 1865 | 392 | 213 | 89 | 137 | 1020 | Fail | N/A |
| | | | $w_{\mathrm{sqrt}}$ | | 3666 | 1955 | 398 | 200 | 58 | 40 | 33 | 32 | 137 |
| | | | $w_{\mathrm{min}}$ | | 3928 | 2034 | 366 | 202 | 55 | 34 | 29 | 24 | 20 |
| | | | $w_{\mathrm{max}}$ | | 3696 | 2000 | 418 | 229 | 62 | 44 | 36 | 34 | Fail |
| dep. | 4 | | $w_{\mathrm{ord}}$ | 21292 | 3898 | 1989 | 422 | 234 | 91 | 117 | 735 | Fail | N/A |
| | | | $w_{\mathrm{sqrt}}$ | | 3977 | 2256 | 394 | 209 | 64 | 44 | 36 | 45 | Fail |
| | | | $w_{\mathrm{min}}$ | | 4196 | 2065 | 412 | 226 | 51 | 35 | 30 | 24 | 20 |
| | | | $w_{\mathrm{max}}$ | | 4360 | 2137 | 504 | 240 | 70 | 46 | 42 | 40 | Fail |
| | 2 | | $w_{\mathrm{ord}}$ | 66020 | 7724 | 6324 | 1033 | 528 | 150 | 145 | 404 | Fail | N/A |
| | | | $w_{\mathrm{sqrt}}$ | | 9458 | 4226 | 1088 | 660 | 180 | 97 | 59 | 72 | Fail |
| | | | $w_{\mathrm{min}}$ | | 8357 | 4363 | 1212 | 484 | 109 | 68 | 54 | 31 | 37 |
| | | | $w_{\mathrm{max}}$ | | 11541 | 7057 | 6794 | 6782 | 3668 | 3124 | 3729 | 7246 | Fail |

Table 3: (BVS) Median of $T_H$, wall-clock time (in seconds) until the chain hit $\gamma^*$ over 50 replicates. Entry with "Fail" indicates that chains never hit $\gamma^*$ in more than half of the replicated datasets.

| | SNR | N | | 1 | 5 | 10 | 50 | 100 | 500 | 1000 | 2000 | 5000 | LBMH |
|---|---|---|---|---|---|---|---|---|---|---|---|---|---|
| ind. | 4 | | $w_{\mathrm{ord}}$ | 1.30 | 0.80 | 0.42 | 0.12 | 0.07 | 0.07 | 0.27 | 6.95 | Fail | N/A |
| | | | $w_{\mathrm{sqrt}}$ | | 0.81 | 0.46 | 0.11 | 0.07 | 0.04 | 0.05 | 0.09 | 0.30 | Fail |
| | | | $w_{\mathrm{min}}$ | | 0.89 | 0.53 | 0.12 | 0.07 | 0.04 | 0.04 | 0.06 | 0.11 | 0.07 |
| | | | $w_{\mathrm{max}}$ | | 0.88 | 0.53 | 0.13 | 0.07 | 0.04 | 0.05 | 0.09 | 0.33 | Fail |
| | 2 | | $w_{\mathrm{ord}}$ | 1.25 | 0.82 | 0.43 | 0.11 | 0.07 | 0.06 | 0.15 | 1.93 | Fail | N/A |
| | | | $w_{\mathrm{sqrt}}$ | | 0.81 | 0.44 | 0.12 | 0.07 | 0.04 | 0.04 | 0.06 | 0.13 | 0.39 |
| | | | $w_{\mathrm{min}}$ | | 1.01 | 0.55 | 0.11 | 0.08 | 0.04 | 0.04 | 0.06 | 0.11 | 0.07 |
| | | | $w_{\mathrm{max}}$ | | 0.92 | 0.51 | 0.13 | 0.08 | 0.05 | 0.05 | 0.06 | 0.13 | Fail |
| dep. | 4 | | $w_{\mathrm{ord}}$ | 1.35 | 0.85 | 0.42 | 0.12 | 0.08 | 0.06 | 0.12 | 1.37 | Fail | N/A |
| | | | $w_{\mathrm{sqrt}}$ | | 0.87 | 0.50 | 0.11 | 0.07 | 0.04 | 0.05 | 0.06 | 0.17 | Fail |
| | | | $w_{\mathrm{min}}$ | | 1.04 | 0.55 | 0.13 | 0.08 | 0.04 | 0.04 | 0.06 | 0.10 | 0.06 |
| | | | $w_{\mathrm{max}}$ | | 1.07 | 0.53 | 0.15 | 0.08 | 0.05 | 0.05 | 0.07 | 0.14 | Fail |
| | 2 | | $w_{\mathrm{ord}}$ | 3.38 | 1.67 | 1.29 | 0.26 | 0.16 | 0.09 | 0.14 | 0.70 | Fail | N/A |
| | | | $w_{\mathrm{sqrt}}$ | | 1.91 | 0.90 | 0.28 | 0.19 | 0.10 | 0.10 | 0.10 | 0.24 | Fail |
| | | | $w_{\mathrm{min}}$ | | 1.89 | 1.01 | 0.34 | 0.16 | 0.07 | 0.07 | 0.10 | 0.13 | 0.09 |
| | | | $w_{\mathrm{max}}$ | | 2.58 | 1.66 | 1.86 | 1.97 | 2.10 | 2.78 | 5.74 | 22.36 | Fail |

In contrast to MTM, LBMH fails to converge to $\gamma^*$ when $h(u) = \sqrt{u}$ or $h(u) = \max\{1, u\}$. It is easier for MTM to escape from such local modes by randomly searching part of its neighborhood to select the proposal. The exhaustive search nature of LBMH makes it difficult to escape from the local mode since some high values of $\pi(y^*)/\pi(y)$, $y^* \in \mathcal{N}(y)$ involved in the denominator makes the acceptance ratio small. This phenomenon disappears when $h(u) = \min\{1, u\}$ is used. In terms of wall-clock hitting time $T_H$, LBMH is not as efficient as MTM with a smaller choice of $N$.

In addition, we also consider the case when SNR $= 0.5$ (very weak SNR) so that the null model $\gamma^* = \mathbf{0}$ receives the highest probability across all simulated datasets. For each replicated dataset, algorithms are randomly initialized with state $\gamma_0$ such that $d_{\mathrm{H}}(\gamma_0, \gamma^*) = 10$ which implies $H = 10$ is the minimum required hitting iteration. Table 4 provides the result similar to Tables 2 and 3, since the posterior distribution is unimodal with the peak at the null model $\gamma^* = \mathbf{0}$ due to the sparsity prior. The median $N$ estimated from Algorithm 2 using $\psi = 0.9$ over 50 replicate datasets is $\hat{N} = 171$ for independent design and is $\hat{N} = 212$ for dependent design.

Table 4: (BVS, very weak SNR $= 0.5$) Median of $H$ and $T_H$ over 50 replicates. Entry with "Fail" indicates that chains never hit $\gamma^*$ in more than half of the replicated datasets.

| | SNR $= 0.5$ | $N$ | 1 | 5 | 10 | 50 | 100 | 500 | 1000 | 2000 | 5000 |
|---|---|---|---|---|---|---|---|---|---|---|---|
| $H$ | indep. | 14358 | $w_{\mathrm{ord}}$ | 2708 | 1309 | 281 | 142 | 50 | 46 | 106 | 8968 |
| | | | $w_{\mathrm{sqrt}}$ | 2788 | 1402 | 285 | 148 | 37 | 20 | 16 | 12 |
| | | | $w_{\mathrm{min}}$ | 2680 | 1481 | 282 | 149 | 34 | 19 | 16 | 12 |
| | | | $w_{\mathrm{max}}$ | 2476 | 1276 | 266 | 131 | 38 | 23 | 17 | 12 |
| | dep. | 12596 | $w_{\mathrm{ord}}$ | 2924 | 1432 | 304 | 142 | 43 | 42 | 104 | Fail |
| | | | $w_{\mathrm{sqrt}}$ | 2776 | 1532 | 278 | 146 | 35 | 21 | 15 | 12 |
| | | | $w_{\mathrm{min}}$ | 2836 | 1354 | 260 | 154 | 34 | 22 | 14 | 11 |
| | | | $w_{\mathrm{max}}$ | 2562 | 1289 | 270 | 140 | 32 | 22 | 15 | 12 |
| $T_H$ | indep. | 0.89 | $w_{\mathrm{ord}}$ | 0.56 | 0.3 | 0.07 | 0.04 | 0.03 | 0.05 | 0.19 | 39.38 |
| | | | $w_{\mathrm{sqrt}}$ | 0.6 | 0.32 | 0.07 | 0.05 | 0.02 | 0.02 | 0.03 | 0.04 |
| | | | $w_{\mathrm{min}}$ | 0.68 | 0.36 | 0.09 | 0.05 | 0.02 | 0.02 | 0.03 | 0.04 |
| | | | $w_{\mathrm{max}}$ | 0.62 | 0.35 | 0.08 | 0.05 | 0.02 | 0.02 | 0.03 | 0.04 |
| | dep. | 0.67 | $w_{\mathrm{ord}}$ | 0.57 | 0.29 | 0.07 | 0.04 | 0.03 | 0.04 | 0.17 | Fail |
| | | | $w_{\mathrm{sqrt}}$ | 0.56 | 0.31 | 0.07 | 0.04 | 0.02 | 0.02 | 0.02 | 0.04 |
| | | | $w_{\mathrm{min}}$ | 0.66 | 0.33 | 0.07 | 0.05 | 0.02 | 0.02 | 0.02 | 0.04 |
| | | | $w_{\mathrm{max}}$ | 0.58 | 0.31 | 0.08 | 0.04 | 0.02 | 0.02 | 0.02 | 0.04 |

## B.2 Details of stochastic block model (SBM)

After marginalizing out $\{Q_{uv}\}_{1 \le u \le v \le K}$, the posterior distribution $\pi(\mathbf{z} \,|\, \boldsymbol{A})$ is written as (see [24, §2.1] and [55, §2.2])

$$\pi(\mathbf{z} \,|\, \boldsymbol{A}) = C \cdot \prod_{1 \le u \le v \le K} B(\kappa_1 + m_{uv}, \kappa_2 + \overline{m}_{uv}) \cdot \mathbb{1}(\mathbf{z} \in S_\alpha), \tag{15}$$

where $B(\kappa_1, \kappa_2) = \Gamma(\kappa_1)\Gamma(\kappa_2)/\Gamma(\kappa_1 + \kappa_2)$ is a beta function, $C$ is a normalizing constant,

$$m_{uv} = \begin{cases} \sum_{i,j} A_{ij} \mathbb{1}(z_i = u, z_j = v) & \text{if } u < v, \\ \sum_{i<j} A_{ij} \mathbb{1}(z_i = u, z_j = u) & \text{if } u = v, \end{cases}$$

is the number of edges between blocks $u$ and $v$, and using the notation $n_u(\mathbf{z}) = \sum_i \mathbb{1}(z_i = u)$,

$$\overline{m}_{uv} = \begin{cases} n_u(\mathbf{z})n_v(\mathbf{z}) - m_{uv} & \text{if } u < v, \\ n_u(\mathbf{z})(n_u(\mathbf{z}) - 1)/2 - m_{uu} & \text{if } u = v, \end{cases}$$

is the number of non-edges between blocks $u$ and $v$. We note that $\pi(\mathbf{z} \,|\, \boldsymbol{A})$ is invariant of a label permutation.

**Data generation.** When $K = 2$, there are two true clusters (blocks) of nodes each with 500 nodes. When $K = 5$, there are five true clusters of nodes each with 200 nodes. We generated a graph from the homogeneous SBM and where within- and cross-community edge connection probabilities are $a$ and $b$ respectively. Specifically, for $K = 2$ we set $(a, b) = (0.222, 0.01)$ and $(a, b) = (0.07, 0.01)$ so that CH $\approx 10$ and CH $\approx 2$, and for $K = 5$ we set $(a, b) = (0.473, 0.01)$ and $(a, b) = (0.13, 0.01)$ so that CH $\approx 10$ and CH $\approx 2$. For each setting, we simulate 50 datasets.

**MCMC setup.** Hyperparameters are specified as $\kappa_1 = \kappa_2 = 1$, and $\alpha = 1000$ so that the size of the feasible set $S_\alpha$ is maximized. For each dataset, we run a chain of $10^5$ iteration for single-try MH, $5 \times 10^4$ iteration for MTM with $N = 5$, and $2 \times 10^4$ iteration for MTM with $N = 5, 10, 50, 100, 500, 1000, 2000, 5000$ using four different weight functions. Algorithms are randomly initialized with state $\mathbf{z}_0$ such that $\tilde{d}_{\mathrm{H}}(\mathbf{z}_0, \mathbf{z}^*) = 400$ which implies $H = 400$ is the minimum required hitting iteration. For each simulated dataset, the true data generated model achieves the highest posterior probability ($\mathbf{z}^* = x^*$). All simulations are performed on a Linux cluster with Intel(R) Xeon(R) Gold 6132 CPU @ 2.60GHz and 96GB memory.

Results from Table 5 show that $H$ decreases roughly by a factor of $N$ until $N = 10$ for locally balanced weight functions, but not for unscaled weight function $w_{\mathrm{ord}}$. Even when $N \ge 50$, MTM with $w_{\mathrm{ord}}$ never converges to the highest probability model, highlighting the necessity of the use of locally balanced weight function for the general model selection problems. When $N$ is very large, the performance of locally balanced weight functions generally deteriorates, which matches with our theoretical findings regarding the rate condition on $N$. Table 6 also suggests that a moderate choice of $N$ (in SBM case, around 10) is beneficial in terms of computation savings. Finally, under different settings of $(K, \mathrm{CH})$, the median $N$ estimated from Algorithm 2 using $\psi = 0.9$ over 50 replicate datasets is $\hat{N} = 15$ for $(K, \mathrm{CH}) = (2, 2)$, $\hat{N} = 8$ for $(2, 10)$, $\hat{N} = 5$ for $(5, 2)$ and $\hat{N} = 4$ for $(5, 10)$.

Comparison of hitting iteration $H$ with LBMH gives an insight similar to the BVS example. When $K = 5$, LBMH often gets stuck at a local mode and never converges to $\mathbf{z}^*$. However when $K = 2$, LBMH performs similarly to MTM with larger choices of $N$. We note that when $K = 2$, the shape of posterior distribution can be significantly different from that of $K = 5$, as the minimax rate and posterior contraction rate analysis are often treated separately when $K = 2$ and $K \ge 3$ [51, 55]. The comparison of wall-clock hitting time $T_H$ also suggests MTM with moderate choice of $N$ is much more efficient.

Table 5: (SBM) Median of $H$, the number of iterations until the chain hit $\mathbf{z}^*$ over 50 replicates. Entry with "Fail" indicates that chains never hit $\mathbf{z}^*$ in more than half of the replicated datasets.

| | CH | | N | 1 | 5 | 10 | 50 | 100 | 500 | 1000 | 2000 | 5000 | LBMH |
|---|---|---|---|---|---|---|---|---|---|---|---|---|---|---|
| $K=2$ | $\approx 10$ | $w_{\mathrm{ord}}$ | | 11572 | 2495 | 5542 | Fail | Fail | Fail | Fail | Fail | Fail | N/A |
| | | $w_{\mathrm{sqrt}}$ | | | 1603 | 1136 | 692 | 644 | 602 | 610 | 637 | 684 | 740 |
| | | $w_{\mathrm{min}}$ | | | 1558 | 974 | 544 | 493 | 444 | 434 | 431 | 424 | 418 |
| | | $w_{\mathrm{max}}$ | | | 1657 | 1142 | 762 | 708 | 726 | 830 | 1224 | 2842 | 6364 |
| | $\approx 2$ | $w_{\mathrm{ord}}$ | | 13343 | 2722 | 3484 | Fail | Fail | Fail | Fail | Fail | Fail | N/A |
| | | $w_{\mathrm{sqrt}}$ | | | 1948 | 1432 | 874 | 818 | 808 | 703 | 700 | 728 | 682 |
| | | $w_{\mathrm{min}}$ | | | 2244 | 1400 | 944 | 911 | 820 | 882 | 812 | 890 | 866 |
| | | $w_{\mathrm{max}}$ | | | 1916 | 1354 | 851 | 774 | 701 | 696 | 681 | 680 | 709 |
| $K=5$ | $\approx 10$ | $w_{\mathrm{ord}}$ | | 25328 | 5852 | 4614 | Fail | Fail | Fail | Fail | Fail | Fail | N/A |
| | | $w_{\mathrm{sqrt}}$ | | | 5400 | 3008 | 1210 | 987 | 858 | 874 | 992 | 1564 | Fail |
| | | $w_{\mathrm{min}}$ | | | 5376 | 2695 | 1127 | 907 | 719 | 690 | 675 | 660 | Fail |
| | | $w_{\mathrm{max}}$ | | | 5230 | 2977 | 1184 | 1022 | 905 | 960 | 1286 | 3895 | Fail |
| | $\approx 2$ | $w_{\mathrm{ord}}$ | | 25883 | 6388 | 4422 | Fail | Fail | Fail | Fail | Fail | Fail | N/A |
| | | $w_{\mathrm{sqrt}}$ | | | 5552 | 2885 | 1067 | 805 | 628 | 574 | 542 | 506 | Fail |
| | | $w_{\mathrm{min}}$ | | | 5426 | 3056 | 1168 | 966 | 802 | 775 | 752 | 740 | 1517 |
| | | $w_{\mathrm{max}}$ | | | 5245 | 2904 | 1100 | 882 | 703 | 654 | 630 | 614 | Fail |

Table 6: (SBM) Median of $T_H$, wall-clock time (in seconds) until the chain hit $\mathbf{z}^*$ over 50 replicates. Entry with "Fail" indicates that chains never hit $\mathbf{z}^*$ in more than half of the replicated datasets.

| | CH | | N | 1 | 5 | 10 | 50 | 100 | 500 | 1000 | 2000 | 5000 | LBMH |
|---|---|---|---|---|---|---|---|---|---|---|---|---|---|---|
| $K=2$ | $\approx 10$ | $w_{\mathrm{ord}}$ | | 0.73 | 0.50 | 1.32 | Fail | Fail | Fail | Fail | Fail | Fail | N/A |
| | | $w_{\mathrm{sqrt}}$ | | | 0.33 | 0.27 | 0.39 | 0.62 | 1.64 | 2.75 | 5.04 | 13.49 | 1.95 |
| | | $w_{\mathrm{min}}$ | | | 0.41 | 0.29 | 0.36 | 0.53 | 1.29 | 2.03 | 3.52 | 8.27 | 1.09 |
| | | $w_{\mathrm{max}}$ | | | 0.42 | 0.35 | 0.49 | 0.75 | 2.05 | 3.75 | 9.90 | 54.05 | 16.13 |
| | $\approx 2$ | $w_{\mathrm{ord}}$ | | 0.82 | 0.54 | 0.83 | Fail | Fail | Fail | Fail | Fail | Fail | N/A |
| | | $w_{\mathrm{sqrt}}$ | | | 0.4 | 0.35 | 0.48 | 0.75 | 1.49 | 2.15 | 3.78 | 9.30 | 1.28 |
| | | $w_{\mathrm{min}}$ | | | 0.56 | 0.41 | 0.57 | 0.89 | 1.55 | 2.62 | 4.31 | 11.26 | 1.60 |
| | | $w_{\mathrm{max}}$ | | | 0.47 | 0.39 | 0.51 | 0.75 | 1.29 | 2.07 | 3.57 | 8.35 | 1.35 |
| $K=5$ | $\approx 10$ | $w_{\mathrm{ord}}$ | | 1.97 | 1.35 | 1.39 | Fail | Fail | Fail | Fail | Fail | Fail | N/A |
| | | $w_{\mathrm{sqrt}}$ | | | 1.25 | 0.92 | 0.97 | 1.38 | 4.01 | 7.36 | 15.71 | 60.95 | Fail |
| | | $w_{\mathrm{min}}$ | | | 2.27 | 1.50 | 1.31 | 1.78 | 3.97 | 6.69 | 12.30 | 29.76 | Fail |
| | | $w_{\mathrm{max}}$ | | | 2.21 | 1.52 | 1.39 | 2.00 | 4.94 | 9.31 | 23.38 | 174.89 | Fail |
| | $\approx 2$ | $w_{\mathrm{ord}}$ | | 1.88 | 1.41 | 1.24 | Fail | Fail | Fail | Fail | Fail | Fail | N/A |
| | | $w_{\mathrm{sqrt}}$ | | | 1.22 | 0.84 | 0.81 | 1.05 | 2.31 | 3.84 | 6.89 | 15.90 | Fail |
| | | $w_{\mathrm{min}}$ | | | 1.35 | 0.97 | 0.91 | 1.31 | 2.99 | 5.17 | 9.61 | 23.64 | 6.52 |
| | | $w_{\mathrm{max}}$ | | | 1.33 | 0.89 | 0.83 | 1.16 | 2.61 | 4.36 | 7.93 | 19.18 | Fail |

## B.3 Spatial clustering model (SCM)

We consider a spatial clustering problem for a given set of spatial locations $\mathcal{S} = \{\mathbf{s}_1, \ldots, \mathbf{s}_p\} \subset \mathbb{R}^2$ where the responses $\zeta(\mathbf{s}_i)$'s are observed. The goal of an SCM is to identify a spatially contiguous partition on $\mathcal{S}$, denoted by $\mathcal{P} = \{\mathcal{S}_1, \ldots, \mathcal{S}_K\}$, where $\mathcal{S}_j$'s are disjoint subsets of $\mathcal{S}$ whose union is $\mathcal{S}$, such that the responses within a cluster $\{\zeta(\mathbf{s}) : \mathbf{s} \in \mathcal{S}_j\}$ are identically distributed and have different means across clusters.

We follow [28] to adopt a probabilistic model for $\mathcal{P}$ that utilizes a spanning tree graph $\mathcal{T}$ on $\mathcal{S}$ (with $p$ vertices and $p-1$ edges) as a "spatial order" of $\mathcal{S}$. The spanning tree $\mathcal{T}$ is chosen in a way that two locations connected by an edge are spatially proximate to each other. A partition $\mathcal{P}$ with $K$ clusters can be defined by removing $K-1$ edges from $\mathcal{T}$. Specifically, the SCM we consider can be written as

$$\zeta(\mathbf{s}_i)|\{\mu(\mathbf{s}_i)\}, \mathcal{P}, K, \sigma^2 \stackrel{\text{ind}}{\sim} \mathsf{N}(\mu_j, \sigma^2), \quad \text{with } \mu(\mathbf{s}_i) = \mu_j \text{ if } \mathbf{s}_i \in \mathcal{S}_j, \text{ for } i \in [p], \quad (16)$$

$$\mu_j|\mathcal{P}, K, \sigma^2 \stackrel{\text{iid}}{\sim} \mathsf{N}(0, \lambda^{-1}\sigma^2), \quad \text{for } j \in [K],$$

$$\sigma^2 \sim \mathsf{InvGamma}(a_0/2, b_0/2),$$

$$\pi(\mathcal{P}|K) \propto \mathbb{1}\{\mathcal{P} \text{ can be obtained by removing } K-1 \text{ edges from } \mathcal{T}\},$$

$$\pi(K) \propto (1-c_0)^K, \quad K = 1, \ldots, p,$$

where $a_0 > 0$, $b_0 > 0$, $0 \le c_0 < 1$, and $\lambda > 0$ are hyperparameters. See Figure 6a for an example of partition $\mathcal{P}$ obtained by cutting edges from the spanning tree $\mathcal{T}$.

Thanks to the conjugate priors, $\mu_j$'s and $\sigma^2$ can be analytically marginalized out, and hence the inference problem boils down to drawing samples from the (discrete) posterior distribution $\pi(\mathcal{P} \,|\, \text{data})$. Although [28] considered random spanning trees by assigning a prior distribution on $\mathcal{T}$, we stress that the main focus of this paper is the mixing time analysis on the posterior distribution of $\mathcal{P}$. Thus, following [26], we fix $\mathcal{T}$ as the Euclidean minimum spanning tree in a Delaunay triangulation graph on $\mathcal{S}$, otherwise we can only sample from the conditional distribution $\pi(\mathcal{P} \,|\, \mathcal{T}, \text{data})$ which complicates the mixing time analysis of our target distribution $\pi(\mathcal{P} \,|\, \text{data})$.

We consider the following proposal:

$$\mathbf{K}_{\text{RW}}(\mathcal{P}, \mathcal{P}') = 1/(p-1)\mathbb{1}_{\mathcal{N}_b(\mathcal{P}) \cup \mathcal{N}_d(\mathcal{P})}(\mathcal{P}'),$$

where $\mathcal{N}_b(\mathcal{P})$ is the set of all possible partitions obtained by splitting a cluster in $\mathcal{P}$ into two clusters by selecting a cut-edge of $\mathcal{T}$ and $\mathcal{N}_d(\mathcal{P})$ is the set of all possible partitions obtained by merging two neighboring (with respect to $\mathcal{T}$) clusters in $\mathcal{P}$. See [23, 28] for detailed discussion on how to perform an appropriate split or merge on $\mathcal{P}$ given a spanning tree $\mathcal{T}$.

**Data generation**. We generate $p = 1000$ uniform locations $\mathbf{s}_i \stackrel{\text{iid}}{\sim} \mathsf{Unif}([0,1]^2), i \in [p]$ and specify the true means $\{\mu(\mathbf{s}_i)\}$ as in Figure 6a. Responses are generated according to (16) with $\sigma = \sqrt{\text{Var}(\mu(\mathbf{s}))}/\text{SNR}$ and we simulate 50 replicate datasets under $\text{SNR} \in \{3, 10\}$ respectively.

**MCMC setup**. Following [28], we initialize the chain using the estimates from the spatially clustered coefficient model of [26]. However, this initialization does not guarantee the same minimum number of iterations required to hit the true partition $\mathcal{P}^{\text{true}}$ for different replicate datasets. For a fair comparison, throughout this subsection, we redefine $H$ as the number of *extra* iterations until hit, which is the iterations until hit minus the minimum number of iterations required to reach $\mathcal{P}^{\text{true}}$. Hyperparameters are specified as $a_0 = b_0 = 1$, $c_0 = 0.5$, and $\lambda = 0.01$. We consider the number of trials $N \in \{5, 10, 100, 500, 1000\}$. For each replicate dataset, we run a chain of $10,000$ iterations for each MTM specification and a chain of $30,000$ for standard single-try MH. All simulation studies are performed on a Linux cluster with Intel(R) Xeon(R) Gold 6132 CPU @ 2.60GHz and 96GB memory.

Table 7 summarizes $H$ and $T_H$ of various weight functions and numbers of trials $N$. The distributions of $H$ for the setting of $\text{SNR} = 10$ are provided in Figure 6b. The results from single-try MH are also included as a baseline. When $\text{SNR} = 10$, the proposed locally balanced weight functions, especially $w_{\text{sqrt}}$ and $w_{\text{min}}$, considerably outperform the ordinary weight function $w_{\text{ord}}$ and the single-try MH, in the sense that the proposed ones can reach $\mathcal{P}^{\text{true}}$ by much fewer iterations when $N \in \{100, 500, 1000\}$. In contrast, the performance of $w_{\text{ord}}$ deteriorates when $N \ge 100$ and it fails to reach $\mathcal{P}^{\text{true}}$ when $N = 500$ or $1000$. For the proposed weight functions, the wall-clock time until hit $T_H$ is minimized when $N = 100$, since the benefit of having fewer iterations until hit is offset by the computational cost of extra trials when $N$ is large.

When SNR = 3, the chains never visit the true partition, possibly because $\mathcal{P}^{\text{true}}$ does not lead to the highest posterior probability. In this case, we redefine $H$ and $T_H$ to be the number of extra iterations and the wall-clock time, respectively, to reach the 0.99 Rand index neighborhood of $\mathcal{P}^{\text{true}}$, defined as

$$\mathcal{N}_{\text{Rand}}(\mathcal{P}^{\text{true}}) \coloneqq \{\mathcal{P} : \text{Rand}(\mathcal{P}, \mathcal{P}^{\text{true}}) \geq 0.99\},$$

where $\text{Rand}(\cdot, \cdot)$ is the Rand index [39] measuring the proportion of agreements between two partitions. The findings on $H$ and $T_H$ for reaching $\mathcal{N}_{\text{Rand}}(\mathcal{P}^{\text{true}})$ are similar to the ones when SNR = 10.

Finally, the median $N$ estimated from Algorithm 2 using $\psi = 0.9$ over 50 replicate datasets is $\hat{N} = 13$ when SNR = 10 and $\hat{N} = 21$ when SNR = 3.

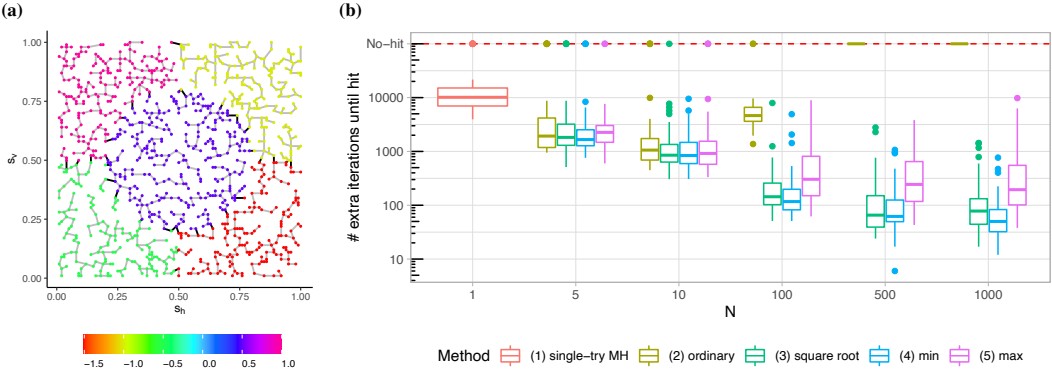

Figure 6: (a) True $\mu(\mathbf{s})$ and the Euclidean minimum spanning tree $\mathcal{T}$ on $\mathcal{S}$. Edges that should be removed in the true partition are marked in black. (b) Boxplot of numbers of extra iterations until hit for SCM under different numbers of trials and weight functions when SNR = 10.

Table 7: (SCM) Median of $H$ and $T_H$ (see text for definition; $T_H$ in seconds) over 50 replicates. Entry with "Fail" indicates that chains never hit the target state in more than half of the replicated datasets.

|  |  | $N$ | 1 | 5 | 10 | 100 | 500 | 1000 |
|---|---|---|---|---|---|---|---|---|
| $H$ | SNR = 10 | $w_{\text{ord}}$ | 10124 | 1933 | 1058 | 4638 | Fail | Fail |
|  |  | $w_{\text{sqrt}}$ |  | 1822 | 851 | 144 | 66 | 78 |
|  |  | $w_{\text{min}}$ |  | 1661 | 840 | 117 | 62 | 50 |
|  |  | $w_{\text{max}}$ |  | 2260 | 918 | 304 | 244 | 195 |
|  | SNR = 3 | $w_{\text{ord}}$ | 23950 | 5083 | 2958 | 1046 | Fail | Fail |
|  |  | $w_{\text{sqrt}}$ |  | 2860 | 2064 | 235 | 84 | 126 |
|  |  | $w_{\text{min}}$ |  | 4443 | 2545 | 234 | 236 | 121 |
|  |  | $w_{\text{max}}$ |  | 6470 | 2810 | 394 | 462 | 301 |
| $T_H$ | SNR = 10 | $w_{\text{ord}}$ | 906.72 | 164.28 | 90.75 | 1291.13 | Fail | Fail |
|  |  | $w_{\text{sqrt}}$ |  | 149.48 | 74.15 | 45.98 | 114.68 | 253.31 |
|  |  | $w_{\text{min}}$ |  | 182.53 | 94.21 | 42.74 | 117.88 | 206.83 |
|  |  | $w_{\text{max}}$ |  | 235.46 | 96.17 | 85.17 | 274.57 | 465.76 |
|  | SNR = 3 | $w_{\text{ord}}$ | 2058.61 | 524.33 | 318.72 | 245.42 | Fail | Fail |
|  |  | $w_{\text{sqrt}}$ |  | 330.36 | 229.80 | 62.23 | 113.15 | 299.45 |
|  |  | $w_{\text{min}}$ |  | 486.18 | 264.62 | 58.71 | 233.03 | 288.35 |
|  |  | $w_{\text{max}}$ |  | 721.71 | 303.91 | 97.47 | 438.06 | 628.02 |

## B.4 MTM algorithm on multimodal target distributions

In this section, we analyze the performance of the MTM algorithm with different choices of weight functions and $N$ on the multimodal target distribution. Following [54], we generate a multimodal dataset in the context of a Bayesian variable selection problem.

**Data generation**. We let sample size $n = 1000$ and number of variables $p = 5000$. Each row of design matrix is independently sampled from $\boldsymbol{x}_i \overset{\text{iid}}{\sim} \mathsf{N}(0, \Sigma)$ for $i = 1, \ldots, n$ where $\Sigma = \text{diag}(\Sigma_{20}, \ldots, \Sigma_{20})$ is block-diagonal. Each block $\Sigma_{20}$ has dimension $20 \times 20$, and $(\Sigma_{20})_{jk} = \exp(-|j - k|/3)$. We generate true coefficient $\boldsymbol{\beta}^{\text{true}}$ by first sampling 100 indices $j_1, \ldots, j_{100}$ uniformly at random (without replacement) from $[p]$ and let $\beta_{j_\ell} \overset{\text{iid}}{\sim} \mathsf{N}(0, \sigma_\beta^2)$ for $\ell = 1, \ldots, 100$, and $\beta_k = 0$ if $k \notin \{j_1, \ldots, j_{100}\}$. Then the response vector $\boldsymbol{y}$ is generated from $\boldsymbol{y} \sim \mathsf{N}(\boldsymbol{X}\boldsymbol{\beta}^{\text{true}}, \mathbf{I}_n)$. We consider three settings of $\sigma_\beta = 0.3, 0.4, 0.5$ to simulate the coefficients and data. For each setting, we simulate 20 datasets.

**MCMC setup**. We use the same BVS model described in Section 4. Hyperparameters are specified as $\mathscr{G} = p = 5000$, $\kappa = 1$, and $s_{\max} = 100$. For each dataset, we run a chain of 10,000 iteration for MTM with $N = 50, 100, 500, 1000, 2000, 5000$ using four different weight functions. The first 2000 iterations are discarded since the behavior of the chain (e.g. acceptance ratio) during the burn-in stage may be different from the behavior of the chain which entered stationarity; see also Figure 7. for trace plots. Algorithms are all initialized with null model $\gamma_0 = \mathbf{0}$. All simulation studies are performed on a Linux cluster with Intel(R) Xeon(R) Gold 6132 CPU @ 2.60GHz and 96GB memory.

Since the target distribution is no longer unimodal, hitting iteration $H$ and wall-clock hitting time $T_H$ are not appropriate metrics to compare mixing performance. Instead, we use three different metrics to evaluate the quality of the mixing: 1) acceptance ratio, 2) the number of unique states visited by the chain, denoted by #(unique $\gamma$), and 3) ESS/Time, where ESS is the effective sample size calculated from the hamming distances $d_{\text{H}}(\hat{\gamma}_{\max}, \gamma_t)$, $t = 2001, \ldots, 10000$ from the maximum posterior state $\hat{\gamma}_{max}$ found in a chain, Time is wall-clock time usage, measured in seconds. The results are summarized in Table 8.

Figure 7: Examples of MTM trace plot using different weight functions and the number of trials $N$. (Left) Simulated data with $\sigma_\beta = 0.3$, (Right) Simulated data with $\sigma_\beta = 0.5$. Each row corresponds to $N = 50, 500, 5000$. All chains are initialized at the null model $\gamma_0 = \mathbf{0}$.

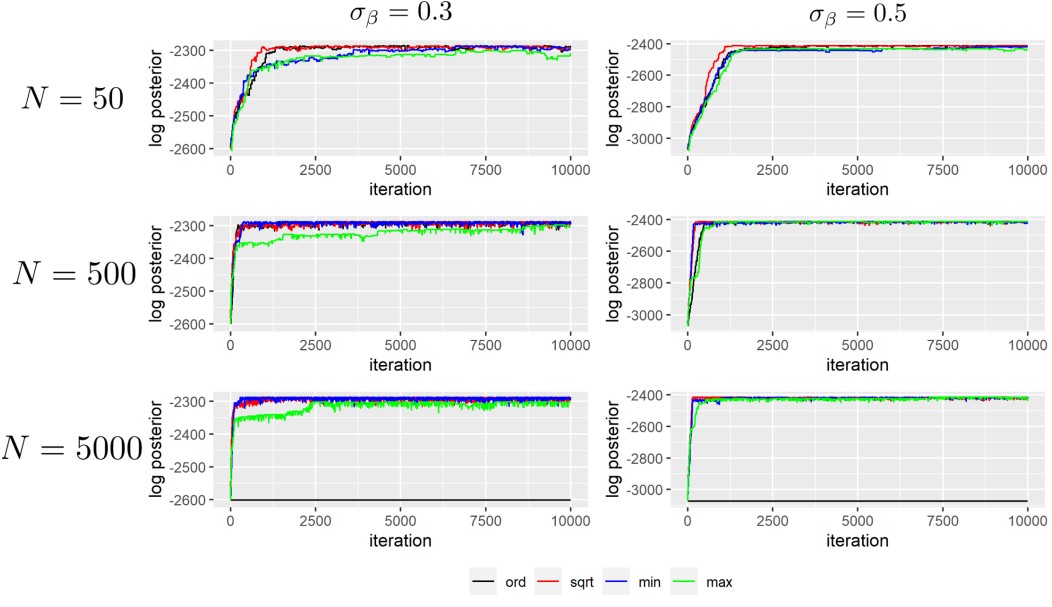

Figure 7 shows that as $N$ increases, chains generally move faster towards the high posterior states which suggests that discarding the first 2000 samples is reasonable. One exception is when $N = 5000$,

Table 8: Multimodal posterior simulation results based on the acceptance rate, the number of unique states visited by a chain, and the effective sample size divided by the running time. All statistics are based on chains with a length of 8000 (2000 burn-in), and are averaged over 20 datasets.

| | | $N$ | 50 | 100 | 500 | 1000 | 2000 | 5000 |
|---|---|---|---|---|---|---|---|---|
| $\sigma_\beta = 0.3$ | Acc. Rate | $w_{\mathrm{ord}}$ | 0.041 | 0.076 | 0.290 | 0.436 | 0.509 | 0.051 |
| | | $w_{\mathrm{sqrt}}$ | 0.039 | 0.069 | 0.191 | 0.257 | 0.314 | 0.385 |
| | | $w_{\min}$ | 0.042 | 0.075 | 0.299 | 0.430 | 0.576 | 0.717 |
| | | $w_{\max}$ | 0.008 | 0.011 | 0.021 | 0.031 | 0.043 | 0.067 |
| | #(unique $\gamma$) | $w_{\mathrm{ord}}$ | 233.4 | 425.5 | 1398.3 | 2052.6 | 2130.8 | 161.8 |
| | | $w_{\mathrm{sqrt}}$ | 240.2 | 385.1 | 993.2 | 1314.7 | 1549.6 | 1885.2 |
| | | $w_{\min}$ | 255.1 | 408.4 | 1538.8 | 2056.4 | 2684.8 | 3168.8 |
| | | $w_{\max}$ | 53.0 | 71.0 | 118.1 | 158.9 | 211.8 | 311.0 |
| | ESS/Time | $w_{\mathrm{ord}}$ | 7.04 | 8.71 | 9.32 | 6.12 | 2.27 | 0.02 |
| | | $w_{\mathrm{sqrt}}$ | 5.11 | 8.08 | 7.14 | 4.62 | 2.48 | 1.14 |
| | | $w_{\min}$ | 5.35 | 7.11 | 9.38 | 7.17 | 4.20 | 1.94 |
| | | $w_{\max}$ | 1.20 | 1.12 | 0.49 | 0.41 | 0.13 | 0.06 |
| $\sigma_\beta = 0.4$ | Acc. Rate | $w_{\mathrm{ord}}$ | 0.036 | 0.069 | 0.260 | 0.389 | 0.417 | 0.017 |
| | | $w_{\mathrm{sqrt}}$ | 0.034 | 0.057 | 0.172 | 0.233 | 0.285 | 0.350 |
| | | $w_{\min}$ | 0.036 | 0.064 | 0.261 | 0.404 | 0.541 | 0.674 |
| | | $w_{\max}$ | 0.007 | 0.009 | 0.018 | 0.026 | 0.040 | 0.059 |
| | #(unique $\gamma$) | $w_{\mathrm{ord}}$ | 211.4 | 392.6 | 1302.9 | 1782.8 | 1852.1 | 72.4 |
| | | $w_{\mathrm{sqrt}}$ | 205.8 | 316.1 | 932.5 | 1277.8 | 1470.2 | 1767.4 |
| | | $w_{\min}$ | 219.8 | 361.6 | 1310.4 | 1919.6 | 2565.6 | 3067.3 |
| | | $w_{\max}$ | 50.2 | 57.4 | 103.6 | 135.3 | 197.6 | 279.4 |
| | ESS/Time | $w_{\mathrm{ord}}$ | 6.13 | 8.06 | 6.31 | 6.58 | 0.52 | 0.03 |
| | | $w_{\mathrm{sqrt}}$ | 4.79 | 6.81 | 6.06 | 4.04 | 2.10 | 1.09 |
| | | $w_{\min}$ | 5.42 | 7.73 | 7.26 | 5.07 | 3.83 | 1.87 |
| | | $w_{\max}$ | 1.04 | 0.80 | 0.35 | 0.23 | 0.11 | 0.05 |
| $\sigma_\beta = 0.5$ | Acc. Rate | $w_{\mathrm{ord}}$ | 0.028 | 0.051 | 0.209 | 0.337 | 0.263 | 0.001 |
| | | $w_{\mathrm{sqrt}}$ | 0.024 | 0.046 | 0.138 | 0.190 | 0.242 | 0.306 |
| | | $w_{\min}$ | 0.026 | 0.055 | 0.209 | 0.337 | 0.481 | 0.633 |
| | | $w_{\max}$ | 0.005 | 0.007 | 0.015 | 0.024 | 0.036 | 0.056 |
| | #(unique $\gamma$) | $w_{\mathrm{ord}}$ | 146.2 | 261.6 | 924.7 | 1382.0 | 976.5 | 5.6 |
| | | $w_{\mathrm{sqrt}}$ | 132.2 | 246.4 | 648.8 | 877.4 | 1054.0 | 1323.6 |
| | | $w_{\min}$ | 138.2 | 291.0 | 958.2 | 1445.6 | 2001.9 | 2444.1 |
| | | $w_{\max}$ | 37.4 | 48.5 | 83.8 | 124.6 | 177.7 | 257.7 |
| | ESS/Time | $w_{\mathrm{ord}}$ | 5.93 | 9.05 | 9.85 | 7.77 | 0.32 | 0.01 |
| | | $w_{\mathrm{sqrt}}$ | 6.75 | 6.83 | 7.33 | 5.05 | 3.27 | 1.17 |
| | | $w_{\min}$ | 6.73 | 6.45 | 8.73 | 5.15 | 5.41 | 2.54 |
| | | $w_{\max}$ | 1.97 | 1.00 | 0.53 | 0.39 | 0.15 | 0.07 |

the chain using ordinary weight function barely moves to another state, whereas the chain using locally balanced weight functions exhibits better mixing properties than $N = 50$ and $N = 500$.

Finally, Table 8 summarizes the multimodal simulation results. Similar to the previous unimodal results, the performance of $w_{\mathrm{ord}}$ deteriorates as $N$ being large, especially when $\sigma_\beta$ is large. When $N$ is moderate, the weighting functions $w_{\mathrm{ord}}$ and $w_{\min}$ has a better mixing property than $w_{\mathrm{sqrt}}$, while $w_{\max}$ being the worst. The inferior performance of $w_{\max}$ is also observed at Table 2. If the current state $x$ is one of the local modes, we may have $\max\{1, \pi(x^\star)/\pi(y)\} \gg \max\{1, \pi(y)/\pi(x)\}$ where $y$ is the proposed state, and $x^\star$ is one of the trials of $y$ in Step 3 of Algorithm 1, which makes the acceptance probability very small so that the chain is stuck at $x$ when using $w_{\max}$. On the other hand, the magnitude of the difference between $h(\pi(x^\star)/\pi(y))$ and $h(\pi(y)/\pi(x))$ will be reduced if we use $w_{\mathrm{sqrt}}, w_{\min}$ so that the chain can traverse among local modes. To summarize, the simulation results suggest that $w_{\min}$ would be the best choice of the weight function if the multimodality exists, since it not only traverses the multimodal posterior efficiently, but also is robust to large $N$. We envision that there are a number of ways to improve the mixing under the multimodal posterior by combining it with techniques such as annealing or tempering [9].

## C  Real data applications

### C.1  GWAS dataset for Bayesian variable selection

We consider a genome-wide association study (GWAS) dataset on glaucoma studied in [54] with sample size $n = 5418$ and number of genetic variants $p = 7255$. The response variable $\mathbf{y} \in \mathbb{R}^{5418}$ is the standardized cut-to-disk ratio measurements averaged over two eyes. We use the BVS model described in Section 4.1, with hyperparameters $\mathscr{G} = 100$ and $\kappa = 0.8$. Since the "true" state is not available, we compare the acceptance rate and the number of unique states visited, averaging over 5 chains. From Table 9, it is clear that the performance of $w_{\mathrm{ord}}$ deteriorates significantly as $N$ grows whereas $w_{\mathrm{sqrt}}, w_{\min}, w_{\max}$ does not. We also report the posterior inclusion probabilities of the top 10 genetic variants in Tables 12, 13, 14, 15, 16, and 17 in Appendix E. All results generally agree with the result of [54], except that when we use $w_{\mathrm{ord}}$ with $N = 5000$, the chain is stuck at local modes and fails to find the significant genetic variants.

Table 9: GWAS dataset analysis results, averaged over 5 chains with random seeds.

| | $N$ | 50 | 100 | 500 | 1000 | 2000 | 5000 |
| --- | --- | --- | --- | --- | --- | --- | --- |
| | iteration | $10^6$ | $5 \times 10^5$ | $10^5$ | $5 \times 10^4$ | $2 \times 10^4$ | $10^4$ |
| | $w_{\mathrm{ord}}$ | 0.4014 | 0.5081 | 0.1370 | 0.0471 | 0.0259 | 0.0085 |
| Acc. Rate | $w_{\mathrm{sqrt}}$ | 0.3407 | 0.4812 | 0.7571 | 0.8325 | 0.8777 | 0.9251 |
| | $w_{\min}$ | 0.4136 | 0.5851 | 0.8252 | 0.8797 | 0.9138 | 0.9455 |
| | $w_{\max}$ | 0.2335 | 0.3404 | 0.6161 | 0.7199 | 0.7930 | 0.8698 |
| | $w_{\mathrm{ord}}$ | 199442 | 126238 | 6796 | 1172 | 259 | 43 |
| #(unique states) | $w_{\mathrm{sqrt}}$ | 169242 | 119396 | 37481 | 20588 | 8669 | 4563 |
| | $w_{\min}$ | 205459 | 145372 | 40993 | 21848 | 9080 | 4696 |
| | $w_{\max}$ | 115615 | 84124 | 30227 | 17591 | 7717 | 4211 |

### C.2  Single-cell RNA dataset for structure learning

We consider a gene expression dataset on Alzheimer's disease used in [10] with the sample size $n = 1666$ and the number of genes $p = 73$. The goal is to learn the underlying directed acyclic graph (DAG) model among the $p$ genes. Due to acyclicity, each DAG has at least one ordering of the nodes. For example, the ordering for the DAG $a \rightarrow b \leftarrow c$ can be either $(a, c, b)$ or $(c, a, b)$. A popular Bayesian structure learning strategy is to use MCMC sampling to first learn the marginal posterior distribution on the order space and then find one or multiple best DAGs for each sampled ordering.

We use an MTM implementation of the order MCMC sampler proposed in [10], which aims to learn the posterior distribution on the order space $\mathbb{S}^p$, the permutation group on $\{1, \ldots, p\}$. The size of our model space $\mathbb{S}^p$ is equal to $73! \approx 4.5 \times 10^{105}$. For each weight function and each setting, we simulate 30 chains, initialized at $(1, \ldots, 73)$. It is clear from Table 10 that the acceptance probability with ordinary weight function $w_{\mathrm{ord}}$ significantly deteriorates, which is consistent with our theory. We can see this tendency more clearly in the log-posterior trace plots for all weight functions in Figure 8.

Table 10: The single-cell RNA database for Alzheimer's disease analysis results, averaged over 30 chains with random seeds. The number in the parenthesis is the standard error.

| | $N$ | 5 | 50 |
| --- | --- | --- | --- |
| | iteration | $5 \times 10^2$ | $2 \times 10^2$ |
| | $w_{\mathrm{ord}}$ | 0.7187 (0.004) | 0.0012 (0.000) |
| Acc. Rate | $w_{\mathrm{sqrt}}$ | 0.8029 (0.004) | 0.9186 (0.003) |
| | $w_{\min}$ | 0.8329 (0.002) | 0.9506 (0.001) |
| | $w_{\max}$ | 0.6643 (0.006) | 0.6806 (0.008) |
| | $w_{\mathrm{ord}}$ | 361.1 (3.1) | 1.333333 (0.1) |
| #(unique orderings) | $w_{\mathrm{sqrt}}$ | 402.3 (2.2) | 184.3 (0.8) |
| | $w_{\min}$ | 416.7 (1.8) | 191.8 (0.6) |
| | $w_{\max}$ | 332.0 (2.9) | 136.2 (1.6) |

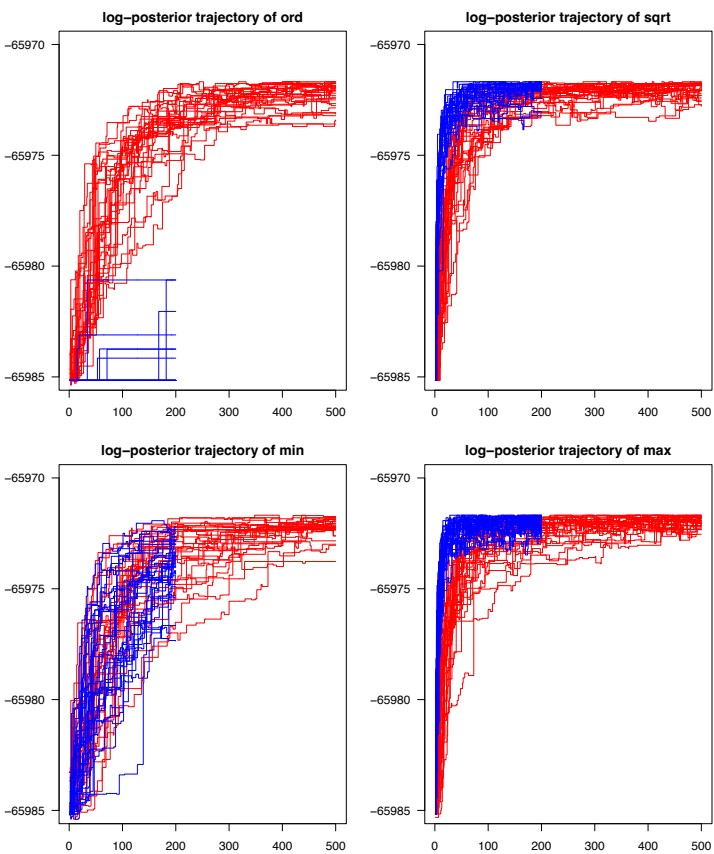

Figure 8: Log-posterior trace plots. Red trajectories indicate MTM with the number of trial $N = 5$ with 500 iterations, and blue trajectories indicate MTM with $N = 50$ with 200 iterations.

# D    Additional discussion

## D.1    On parallelization (vectorization)

As discussed in *Remark 3* in Section 3.1, the overall theoretical computational complexity of the MTM algorithm until the convergence remains the same as a usual MH algorithm. However, MTM enables parallel computations when evaluating $N$ weight functions and hence leads to a significant practical computational gain as evidenced by the reduced wall-clock hitting time reported in Table 1. Under the random walk proposal $\mathbf{K}_{\mathrm{RW}}$, the evaluation of weight functions is equivalent to the evaluation of target distribution at $N$ states $y_1, \ldots, y_N$. Here we clarify that the scope of parallelism we consider is the instruction-level parallelism [41], also called vectorization. Thanks to modern When task-level parallelism, assigning a set of independent tasks in parallel across several processors, is employed to the MTM algorithm within each MCMC iteration, it suffers from communication overhead unless the evaluation of target distributions takes extremely long. Thanks to the optimized linear algebra libraries such as BLAS [7], the easiest way to achieve instruction-level parallelism is to convert the problem of evaluating target distribution at multiple states $\pi(y_1), \ldots, \pi(y_N)$ to a series of matrix multiplication problems.

Here we outline the computational strategy to simultaneously calculate $\pi(y_1), \ldots, \pi(y_N)$ for BVS and SBM. For BVS, since only one variable is added or deleted in the proposal, the Cholesky rank-1 update [17, 44] is utilized to get $\pi(y_1), \ldots, \pi(y_N)$ from $\pi(x)$. To be specific, assume $\gamma'_1 \ldots, \gamma'_N$ are obtained by adding a variable from $\gamma$. By (14), the evaluation of $\pi(\gamma_j \mid \boldsymbol{y})$, $j = 1, \ldots, N$ corresponding to evaluating $\mathrm{SSR}(\gamma_j)$, $j = 1, \ldots, N$ from $\mathrm{SSR}(\gamma)$ saved from the previous iteration. We refer [49, Appendix B] for details of vectorization procedure with Cholesky rank-1 update. For SBM, let $A_i \in \{0, 1\}^p$ be the $i$th column of adjacency matrix and $Z \in \{0, 1\}^{p \times K}$ be one-hot encoded partition matrix such that $Z_{i,k} = 1$ if $z_i = k$ and 0 otherwise. Since a node is assigned to another block one at a time, the calculation of $\pi(y_1) \ldots, \pi(y_N)$ given current state $\pi(x)$ can be done by counting the change of the number of edges between blocks; see (15). Letting $A_J \in \{0, 1\}^{p \times N}$ where column $A_j$ corresponds to the $j$th proposal, the matrix-matrix multiplication $Z^\top A_J$ allows to calculate $\pi(y_1) \ldots, \pi(y_N)$ simultaneously from $\pi(x)$. In addition, if the graph is sparse, then sparse matrix multiplication algorithms can be utilized for further speedup.

## D.2    On state space

Our state space of interest is finite (so discrete), but the proposed locally balanced MTM algorithm is also applicable to continuous state spaces which will be shown shortly. We choose to focus on the discrete case since the theory on continuous state spaces is usually developed under very different frameworks (and likewise, the theory on continuous spaces often cannot be readily applied to discrete ones). Indeed, developing MCMC theory or methodology on discrete spaces is often regarded as more challenging than on continuous ones [48, Section 1], due to the lack of gradient information and a widely accepted theoretical framework supported by statistical theory (for comparison, on continuous spaces, one often assumes log-concavity or asymptotic normality of the target posterior distribution).

To some extent, the proposed MTM method is conceptually similar to MALA (Metropolis adjusted Langevin algorithm) or HMC (Hamiltonian Monte Carlo) on continuous spaces in that MTM evaluates the "gradient" by a random search of neighboring states. This suggests that for continuous-state-space problems where the gradient of log-posterior cannot be easily evaluated (e.g. Bayesian inverse problems and Gaussian process regression models), the proposed MTM method can be quite useful.

We conclude this section with a simulation study that shows the weight function proposed in Proposition 2 can lead to an improved MTM algorithm on continuous spaces. Suppose our target distribution is the 10-dimensional Gaussian distribution $\mathsf{N}(0, \mathbf{I}_{10})$. We set our proposal distribution $q(\cdot \mid x) = \mathsf{N}_{10}(x, 10^{-2}\mathbf{I}_{10})$, initialize the chains at $x_0 = (10, 10, \ldots, 10)$, and run 10,000 iterations for each chain. The result is summarized in Table 11, where for each setting we repeat the simulation 30 times. The advantage of the weight functions considered in Proposition 2 over $w_{\mathrm{ord}}$ is substantial. We present the log-posterior traceplots in Figure 9 and the MCMC sample trajectories in Figure 10.

Table 11: Sampling from 10-d standard Gaussian distribution with 10,000 iterations using MTM. Averaged over 30 chains with random seeds. The number in the parenthesis is the standard error.

| | $N$ | MH ($N=1$) | 10 | 100 | 1,000 |
|---|---|---|---|---|---|
| | $w_{\mathrm{ord}}$ | | 0.7488 (0.002) | 0.2971 (0.012) | 0.0394 (0.008) |
| Acc. Rate | $w_{\mathrm{sqrt}}$ | 0.8604 (0.001) | 0.9667 (0.000) | 0.9897 (0.000) | 0.9967 (0.000) |
| | $w_{\mathrm{min}}$ | | 0.9656 (0.000) | 0.9890 (0.000) | 0.9963 (0.000) |
| | $w_{\mathrm{max}}$ | | 0.9593 (0.000) | 0.9861 (0.000) | 0.9950 (0.000) |

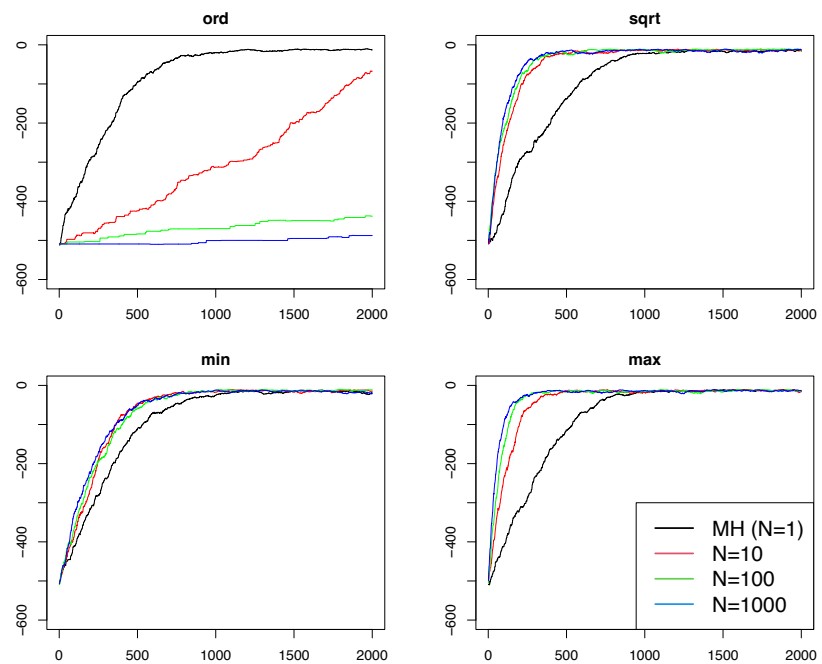

Figure 9: Log-posterior trace plots for 4 different weight functions $w_{\mathrm{ord}}, w_{\mathrm{sqrt}}, w_{\mathrm{min}}$ and $w_{\mathrm{max}}$. Different colors indicate a different number of trials as specified in the legend.

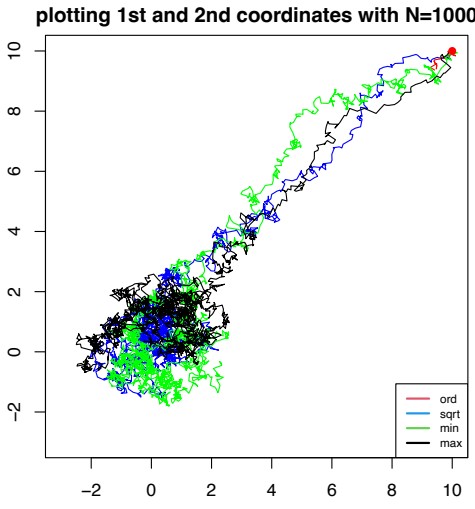

Figure 10: MCMC sample trajectories under 10-dimensional continuous target distribution $\mathsf{N}(\mathbf{0}, \mathbf{I}_{10})$ with different weight functions, initialized at $x_0 = (10, 10, \ldots, 10)$. Note that the chain with the ordinary weight function $w_{\mathrm{ord}}$ gets stuck at its early stage, whereas the chains with the other weight functions move to the region with a high posterior, which supports our claim.

# E   Additional tables

Here we report genetic variants with the top 10 highest posterior inclusion probabilities found from BVS model with MTM algorithm, under the different choices of $N = 50, 100, 500, 1000, 2000,$ 5000 and weight functions $w_{\text{ord}}, w_{\text{sqrt}}, w_{\text{min}}$ and $w_{\text{max}}$.

Table 12: Genetic variants with top 10 posterior inclusion probability (PIP), obtained from MTM algorithm with $N = 50$ and averaged over 5 chains. Blue are genetic variants reported by [54].

|    | $w_{\text{ord}}$ | | $w_{\text{sqrt}}$ | | $w_{\text{min}}$ | | $w_{\text{max}}$ | |
|----|------|-----|------|-----|------|-----|------|-----|
|    | Name | PIP | Name | PIP | Name | PIP | Name | PIP |
| 1  | rs1063192 | 0.987 | rs10483727 | 0.997 | rs1063192 | 0.998 | rs1063192 | 0.982 |
| 2  | rs653178 | 0.979 | rs1063192 | 0.979 | rs653178 | 0.976 | rs10483727 | 0.976 |
| 3  | rs10483727 | 0.972 | rs653178 | 0.978 | rs10483727 | 0.974 | rs653178 | 0.972 |
| 4  | rs2275241 | 0.908 | rs2275241 | 0.914 | rs2275241 | 0.915 | rs2275241 | 0.903 |
| 5  | rs319773 | 0.806 | rs319773 | 0.798 | rs4557053 | 0.802 | rs319773 | 0.827 |
| 6  | rs4557053 | 0.77 | rs4557053 | 0.773 | rs319773 | 0.777 | rs4557053 | 0.752 |
| 7  | rs2369705 | 0.667 | rs2369705 | 0.678 | rs2369705 | 0.671 | rs2369705 | 0.667 |
| 8  | rs10491971 | 0.619 | rs10491971 | 0.639 | rs10491971 | 0.63 | rs10491971 | 0.633 |
| 9  | rs3177954 | 0.598 | rs3177954 | 0.604 | rs3177954 | 0.595 | rs3177954 | 0.602 |
| 10 | rs11087973 | 0.567 | rs11087973 | 0.581 | rs11087973 | 0.576 | rs3843894 | 0.533 |

Table 13: Genetic variants with top 10 posterior inclusion probability (PIP), obtained from MTM algorithm with $N = 100$ and averaged over 5 chains. Blue are genetic variants reported by [54].

|    | $w_{\text{ord}}$ | | $w_{\text{sqrt}}$ | | $w_{\text{min}}$ | | $w_{\text{max}}$ | |
|----|------|-----|------|-----|------|-----|------|-----|
|    | Name | PIP | Name | PIP | Name | PIP | Name | PIP |
| 1  | rs10483727 | 1 | rs10483727 | 0.993 | rs10483727 | 0.988 | rs10483727 | 1 |
| 2  | rs1063192 | 0.996 | rs1063192 | 0.99 | rs653178 | 0.97 | rs1063192 | 0.982 |
| 3  | rs653178 | 0.973 | rs653178 | 0.973 | rs1063192 | 0.953 | rs653178 | 0.975 |
| 4  | rs2275241 | 0.918 | rs2275241 | 0.922 | rs2275241 | 0.914 | rs2275241 | 0.925 |
| 5  | rs319773 | 0.795 | rs4557053 | 0.806 | rs319773 | 0.802 | rs319773 | 0.793 |
| 6  | rs4557053 | 0.795 | rs319773 | 0.78 | rs4557053 | 0.792 | rs4557053 | 0.765 |
| 7  | rs2369705 | 0.673 | rs2369705 | 0.663 | rs2369705 | 0.662 | rs2369705 | 0.63 |
| 8  | rs10491971 | 0.626 | rs10491971 | 0.624 | rs10491971 | 0.617 | rs11087973 | 0.604 |
| 9  | rs3177954 | 0.618 | rs3177954 | 0.62 | rs3177954 | 0.598 | rs10491971 | 0.602 |
| 10 | rs11087973 | 0.583 | rs11087973 | 0.597 | rs11087973 | 0.587 | rs11040978 | 0.563 |

Table 14: Genetic variants with top 10 posterior inclusion probability (PIP), obtained from MTM algorithm with $N = 500$ and averaged over 5 chains. Blue are genetic variants reported by [54].

|    | $w_{\text{ord}}$ | | $w_{\text{sqrt}}$ | | $w_{\text{min}}$ | | $w_{\text{max}}$ | |
|----|------|-----|------|-----|------|-----|------|-----|
|    | Name | PIP | Name | PIP | Name | PIP | Name | PIP |
| 1  | rs1063192 | 1 | rs1063192 | 1 | rs1063192 | 1 | rs10483727 | 1 |
| 2  | rs653178 | 0.992 | rs10483727 | 0.983 | rs10483727 | 1 | rs653178 | 0.925 |
| 3  | rs2275241 | 0.943 | rs653178 | 0.979 | rs653178 | 0.973 | rs2275241 | 0.915 |
| 4  | rs319773 | 0.857 | rs2275241 | 0.922 | rs2275241 | 0.887 | rs319773 | 0.839 |
| 5  | rs10483727 | 0.8 | rs319773 | 0.791 | rs319773 | 0.792 | rs4557053 | 0.828 |
| 6  | rs4557053 | 0.787 | rs4557053 | 0.774 | rs4557053 | 0.791 | rs1063192 | 0.799 |
| 7  | rs10491971 | 0.658 | rs10491971 | 0.67 | rs2369705 | 0.65 | rs2369705 | 0.764 |
| 8  | rs3843894 | 0.579 | rs2369705 | 0.647 | rs3177954 | 0.622 | rs10491971 | 0.757 |
| 9  | rs587409 | 0.553 | rs3177954 | 0.624 | rs10491971 | 0.615 | rs12133371 | 0.617 |
| 10 | rs3177954 | 0.546 | rs11087973 | 0.613 | rs11087973 | 0.561 | rs11087973 | 0.605 |

Table 15: Genetic variants with top 10 posterior inclusion probability (PIP), obtained from MTM algorithm with $N = 1000$ and averaged over 5 chains. Blue are genetic variants reported by [54].

| | $w_{\mathrm{ord}}$ | | $w_{\mathrm{sqrt}}$ | | $w_{\mathrm{min}}$ | | $w_{\mathrm{max}}$ | |
|---|---|---|---|---|---|---|---|---|
| | Name | PIP | Name | PIP | Name | PIP | Name | PIP |
| 1 | rs653178 | 0.959 | rs1063192 | 0.991 | rs1063192 | 1 | rs10483727 | 1 |
| 2 | rs2275241 | 0.952 | rs653178 | 0.982 | rs10483727 | 1 | rs653178 | 0.999 |
| 3 | rs319773 | 0.918 | rs10483727 | 0.939 | rs653178 | 0.994 | rs2275241 | 0.886 |
| 4 | rs10483727 | 0.8 | rs2275241 | 0.927 | rs2275241 | 0.918 | rs2369705 | 0.842 |
| 5 | rs2369705 | 0.737 | rs4557053 | 0.866 | rs319773 | 0.791 | rs319773 | 0.841 |
| 6 | rs10491971 | 0.713 | rs319773 | 0.859 | rs4557053 | 0.779 | rs1063192 | 0.804 |
| 7 | rs4557053 | 0.704 | rs2369705 | 0.651 | rs10491971 | 0.652 | rs3177954 | 0.787 |
| 8 | rs3177954 | 0.688 | rs10491971 | 0.637 | rs2369705 | 0.646 | rs10491971 | 0.768 |
| 9 | rs12133371 | 0.615 | rs3177954 | 0.6 | rs3177954 | 0.613 | rs12133371 | 0.656 |
| 10 | rs1063192 | 0.6 | rs11087973 | 0.574 | rs11087973 | 0.588 | rs3843894 | 0.65 |

Table 16: Genetic variants with top 10 posterior inclusion probability (PIP), obtained from MTM algorithm with $N = 2000$ and averaged over 5 chains. Blue are genetic variants reported by [54].

| | $w_{\mathrm{ord}}$ | | $w_{\mathrm{sqrt}}$ | | $w_{\mathrm{min}}$ | | $w_{\mathrm{max}}$ | |
|---|---|---|---|---|---|---|---|---|
| | Name | PIP | Name | PIP | Name | PIP | Name | PIP |
| 1 | rs2275241 | 1 | rs653178 | 0.983 | rs1063192 | 0.998 | rs1063192 | 1 |
| 2 | rs653178 | 1 | rs10483727 | 0.922 | rs653178 | 0.971 | rs10483727 | 0.997 |
| 3 | rs319773 | 0.819 | rs2275241 | 0.857 | rs2275241 | 0.916 | rs653178 | 0.986 |
| 4 | rs11087973 | 0.805 | rs1063192 | 0.829 | rs319773 | 0.794 | rs2275241 | 0.948 |
| 5 | rs1063192 | 0.8 | rs4557053 | 0.824 | rs4557053 | 0.771 | rs319773 | 0.87 |
| 6 | rs4557053 | 0.755 | rs10491971 | 0.775 | rs10483727 | 0.724 | rs2369705 | 0.675 |
| 7 | rs1460509 | 0.737 | rs319773 | 0.763 | rs10491971 | 0.714 | rs4557053 | 0.634 |
| 8 | rs3843894 | 0.729 | rs2369705 | 0.697 | rs3177954 | 0.672 | rs3177954 | 0.527 |
| 9 | rs3858886 | 0.648 | rs11040978 | 0.651 | rs2369705 | 0.628 | rs2567344 | 0.519 |
| 10 | rs10483727 | 0.61 | rs587409 | 0.612 | rs3843894 | 0.576 | rs11634375 | 0.472 |

Table 17: Genetic variants with top 10 posterior inclusion probability (PIP), obtained from MTM algorithm with $N = 5000$ and averaged over 5 chains. Blue are genetic variants reported by [54].

| | $w_{\mathrm{ord}}$ | | $w_{\mathrm{sqrt}}$ | | $w_{\mathrm{min}}$ | | $w_{\mathrm{max}}$ | |
|---|---|---|---|---|---|---|---|---|
| | Name | PIP | Name | PIP | Name | PIP | Name | PIP |
| 1 | rs2151280 | 0.4 | rs1063192 | 1 | rs1063192 | 1 | rs10483727 | 1 |
| 2 | rs10483727 | 0.367 | rs10483727 | 1 | rs10483727 | 1 | rs653178 | 0.997 |
| 3 | rs12457539 | 0.29 | rs653178 | 0.983 | rs653178 | 0.972 | rs319773 | 0.901 |
| 4 | rs10508818 | 0.255 | rs319773 | 0.804 | rs2275241 | 0.821 | rs3177954 | 0.844 |
| 5 | rs3858886 | 0.231 | rs2369705 | 0.791 | rs4557053 | 0.732 | rs2275241 | 0.668 |
| 6 | rs7995962 | 0.207 | rs2275241 | 0.772 | rs319773 | 0.709 | rs12457539 | 0.649 |
| 7 | rs12125527 | 0.2 | rs3177954 | 0.732 | rs3177954 | 0.662 | rs1063192 | 0.6 |
| 8 | rs2738755 | 0.2 | rs587409 | 0.652 | rs12133371 | 0.593 | rs587409 | 0.6 |
| 9 | rs6661853 | 0.2 | rs10491971 | 0.636 | rs11087973 | 0.569 | rs4924156 | 0.51 |
| 10 | rs9869577 | 0.2 | rs4557053 | 0.611 | rs587409 | 0.565 | rs4236601 | 0.498 |