# OpenReview forum: "Rapidly Mixing Multiple-try Metropolis Algorithms for Model Selection Problems"
_NeurIPS.cc/2022/Conference — NeurIPS 2022 Accept_

### Official Review · Reviewer_x268 · 2022-07-11

**Rating:** 7
**Confidence:** 4
**Soundness:** 3 good
**Presentation:** 3 good
**Contribution:** 3 good

**Summary:**

The authors studied the mixing time of the MTM algorithm.

They propose a new class of weights (or a generalization of the previous weights).

**Questions:**

- I believe that the problem with the number of tries (`` It has been reported with empirical evidence that an increase in the number of trials N does not necessarily result in a corresponding improvement in mixing'') could be associated to the choice of the weights.
Using Importance Sampling (IS) weights this behavior should disappear. Please, discuss it.

- If you have some results results associated to the optimal weights, please remark them.

- In order to complete the state-of-the-art discuss, consider

L. Martino, "A Review of Multiple Try MCMC algorithms for Signal Processing", Digital Signal Processing, Volume 75, Pages: 134-152, 2018.

S. Pandolfi, F. Bartolucci, N.Friel, A generalization of the Multiple-try Metropolis algorithm for Bayesian estimation and model selection, Journal of Machine Learning Research (Workshop and Conference Proceedings Volume 9: AISTATS 2010) 9 (2010) 581–588

and regarding the balancing functions a good reference is

L. Martino, V. Elvira. "Metropolis Sampling", Wiley StatsRef: Statistics Reference Online, 2017.

- The explanations of the models in the numerical examples are extremely synthetic.
Try to give more information if possible.

**Limitations:**

Some parts are not easy to read (maybe less synthesis could help )

**Strengths And Weaknesses:**

The analysis of the mixing time is important and the MTM scheme is a relevant algorithm.
Moreover, the paper is well-written in my opinion.

Since it is a bit mathematical work, some parts are not easy to follow. Also the numerical examples are hard to read.

---

> ### Author Response · Authors · 2022-08-01
> **Response to Reviewer 3 (Reviewer x268)**
>
> Thank you so much for the constructive review. Since there are 4 major overlapped comments raised by multiple reviewers, we created separate “**Response to all**” threads to address these common concerns. We also uploaded ‘**rebuttal.pdf**’ in the supplementary zip file, which contains all of our reviews, to present more detail with an  **easy-to-look presentation**.
>
>
> ### Strengths and Weaknesses ###
>
> **A:** Thank you for the comment. We agree that the main text is very dense due to the page limit. To address your concern that our simulation is too synthetic, we added two real-world data analyses:
> (1) a genome-wide association study dataset for Bayesian variable selection, and (2) a single-cell RNA dataset for  Bayesian structure learning. Please see our preliminary results in **Response to all: Section A**. In summary, the results are consistent with our theory. We plan to make some figures to help understand a path method in revision, which may make it easier to follow.
>
> ### Questions and Limitations ###
>
> > I believe that the problem with the number of tries ("It has been reported with empirical evidence that an increase in the number of trials N does not necessarily result in a corresponding improvement in mixing") could be associated to the choice of the weights. Using Importance Sampling (IS) weights this behavior should disappear. Please, discuss it.
>
> **A:** The problem with large $N$ is indeed associated with the choice of weight functions. Our experiments show that MTM with ordinary weight function $w(y|x) = \pi(y)$ fails to converge when $N$ becomes larger. In the experiments, we used random walk proposal kernel $K_{RW}(x,\cdot)$ where its probabilities are uniform across all $x$. Then, using the weight function of the form $w(y|x) = \pi(y)/K_{RW}(x,y)$ (which resembles the importance sampling weights) is equivalent to the ordinary weight function, which fails as $N$ becomes larger. This phenomenon may change when we choose a different proposal kernel $K(x,\cdot)$ and use weight function of form $w(y|x) = \pi(y)/K(x,y)$. For example, for independent MTM (or MTM-IS) where the proposal kernel does not depend on the current state, this behavior may disappear as described in Section 5 of [Martino2017a]; but such proposals are usually not applicable to model selection problems, and designing an "informed" local proposal kernel on a high-dimensional discrete space is intrinsically difficult and computationally expensive. We will revise the statement "It has been reported with empirical evidence that an increase in the number of trials N does not necessarily result in a corresponding improvement in mixing" to discuss the effect of the choice of weights.
>
> > If you have some results associated to the optimal weights, please remark them.
>
> **A:** Choosing an appropriate weight function is an important problem in MTM. In our main theorem, we require the weight function to have the form of Equation (4) in Proposition 2 with nondecreasing balancing function $h$. We compared three different weight functions satisfying this requirement, and the performance depends on how the posterior landscape looks like. For example with large $N$, in the high SNR in BVS problem $w_{min}$ performed well (lines 308-311) whereas in low SNR in SBM problem, $w_{sqrt}$ and $w_{max}$ performed well (lines 337-341). For the multimodal distribution, a discussion on the weight functions can be found in Appendix lines 271-283. We will highlight these findings in revision.
>
> > In order to complete the state-of-the-art discuss, consider [Martino2018], [Pandolfi2010], and regarding the balancing functions a good reference is [Martino2017b].
>
> **A:** Thank you for the suggestion, we will make sure to cite and discuss them in revision.
>
> > The explanations of the models in the numerical examples are extremely synthetic. Try to give more information if possible.
>
> **A:** Thank you. Due to the page limit, we could not include all the details of our examples in the main text, but we deferred them to the appendix. We will revise the main manuscript to include enough explanations and interpretations of the simulation models. In addition, we have conducted two real-world data analyses. Please see our preliminary results in **Response to all: Section A**. We will make them complete in revision.
>
>
> ### References ###
>
> [Martino2017a] Luca Martino and Francisco Louzada. Issues in the multiple try metropolis mixing. Computational Statistics, 32(1):239–252, 2017.
>
> [Martino2017b] Luca Martino and Victor Elvira. Metropolis sampling. arXiv preprint arXiv:1704.04629, 2017.
>
> [Martino2018]  Luca Martino. A review of multiple try mcmc algorithms for signal processing. Digital Signal
> Processing, 75:134–152, 2018.
>
> [Pandolfi2010] Silvia Pandolfi, Francesco Bartolucci, and Nial Friel. A generalization of the multiple-try
> metropolis algorithm for bayesian estimation and model selection. AISTATS, 2010.

---

### Official Review · Reviewer_CaBA · 2022-07-18

**Rating:** 7
**Confidence:** 3
**Soundness:** 3 good
**Presentation:** 3 good
**Contribution:** 2 fair

**Summary:**

Update: I acknowledge the rebuttal and raise my score to accept given the extensive discussion and provided improvements.

The paper studies multiple try metropolis (MTM) Hastings algorithms, a class of MCMC algorithms for discrete state spaces. A theoretical bound on the mixing time is provided that shows that MTM mixes faster than standard Metropolis Hastings (MH). The results suggests a new class of weight functions that improves upon standard MTM. The results are highlighted empirically in a simulation study using synthetic data, focusing on model (variable) selection problems.


**Questions:**

Does your algorithm apply to discrete state spaces only? Could you make this more explicit, please?
Algorithm 2:
Do you have to run Alg. 2 several times? This looks highly dependent on your initial state x_0 (that might be fare away from regions of high posterior probability).

**Limitations:**

In my understanding, the approach applies to discrete state spaces only and the experimental results lack use of real world data and comparison with other approaches.

**Strengths And Weaknesses:**

The paper provides a theoretical analysis and an extension of existing theory that looks sound. The paper is well written and motivates in detail the perspective taken.
Experiments:
“The trace plot shows that single-try MH reaches the true state γ∗ at around 20,000 iterations, whereas the MTM with N = 10 reaches the true state γ∗ at around 2,000 iterations, smaller by a factor of 10”.
It really looks as if the computation gain is achieved by only shifting computational cost from a longer chain to a shorter chain with more trials. This does not seem too convincing, in particular as parallelisation does not seem to be the main motivation for this work.
The experiment section is also lacking comparison with other methods (such as RJMCMC, for example) to better situate the contribution. I would have also liked to see some real data applications, that is absent from this paper.

---

> ### Author Response · Authors · 2022-08-01
> **Response to Reviewer 2 (Reviewer CaBA)**
>
> Thank you so much for the constructive review. Since there are 4 major overlapped comments raised by multiple reviewers, we created separate “Response to all” threads to address these common concerns. We also uploaded ‘**rebuttal.pdf**’ in the supplementary zip file, which contains all of our reviews, to present more detail with an  **easy-to-look presentation**.
>
> ### Strengths and Weaknesses ###
>
> **A:** Thank you for your constructive comments. We made the most of our effort to resolve all your concerns in the rebuttal. First, we have conducted two real data application analyses (see **Response to all: Section A** above) and comparative simulations with another MCMC method (see **Response to all: Section B**), respectively. We also discuss parallelization in **Response to all: Section C**. Finally, we have commented on the issue of state space in **Response to all: Section D**. Thanks to your feedback, our manuscript will look more comprehensive in revision.
>
> > Does your algorithm apply to discrete state spaces only? Could you make this more explicit, please?
>
> **A:** Our state space of interest is discrete (lines 19, 32, 40, 71, 104, 116, 142), but the locally balanced MTM can be applied to continuous spaces as well. We will make it more explicit in the updated manuscript, including the revision of the abstract. See **Response to all: Section D**, for the detailed comment on state space and the simulation study for the continuous case. We will add the related discussion in revision.
>
> > Algorithm 2: Do you have to run Alg. 2 several times? This looks highly dependent on your initial state $x_0$ (that might be far away from regions of high posterior probability).
>
> **A:** In our current numerical examples, we run Algorithm 2 only once per chain. Yes, it depends on the choice of initial state $x_0$, and it can indeed be far away from the regions of the high posterior probability. The main rationale of algorithm 2 is to guide the number of trials $N$ to comply with Condition (iii) in Theorem 1 (asymptotic upper bound of $N$).
> This corresponds to searching constants $t_1$ and $t_2$ in Theorem 1 (since $t_3,t_4$ are known from the design of the neighborhood relation). As we stated in lines 265-271, running Algorithm 2 at an initial state $x_0$ with low posterior probability provides a simple yet conservative choice of $t_2$ (i.e. small $t_2$) which in turn leads to a small and conserve choice of $N$ (recall Condition (iii) states the asymptotic upper bound). To become even more conservative, one can run Algorithm 2 multiple times with different initial states and choose $N$ as a minimum of those.
>
> ### Limitations ###
>
> > In my understanding, the approach applies to discrete state spaces only and the experimental results lack use of real world data and comparison with other approaches.
>
> **A:** We have addressed both of your concerns.
> First, we added two real-world data analyses: (1) a genome-wide association study dataset for Bayesian variable selection, and (2) a single-cell RNA dataset for  Bayesian structure learning. Please see our preliminary results in **Response to all: Section A**.
> In summary, the results are consistent with our theory.  There are many other real data application scenarios with discrete spaces where the proposed MTM is applicable, such as community detection problems on networks and spatial clustering problems. Second, we added the comparison with the recently proposed locally balanced MH algorithm of [Zanella2020]. Please see **Response to all: Section B**. Last but not least, the proposed algorithm is also applicable to continuous state spaces (but the theory on continuous spaces would be a very different problem). In **Response to all: Section D**, we present a new simulation result to empirically show the improved performance of locally balanced MTM for a continuous-state-space model over conventional MTM. Thank you very much for the comments, which helped improve our paper significantly.
>
> ### Reference ###
>
> [Zanella2020] Giacomo Zanella. Informed proposals for local mcmc in discrete spaces. Journal of the American Statistical Association, 115(530):852–865, 2020.

---

### Official Review · Reviewer_8J7v · 2022-07-18

**Rating:** 7
**Confidence:** 3
**Soundness:** 3 good
**Presentation:** 3 good
**Contribution:** 3 good

**Summary:**

EDIT: satisfied by the rebuttal and the proposed modifications by the authors, I decided to increase my grade by 1 point.

The paper studies multiple-try Metropolis for discrete state spaces, a variant of MH where one draws several times independently from the same proposal at each iteration. The main result is a bound on the mixing time, with an additional algorithmic degree of freedom (the balancing function) compared to previous work. An algorithm for selecting the number of trials in each MCMC iteration is also given. Supporting experiments are shown in two Bayesian model selection tasks.


**Questions:**

## Questions to discuss
Here are a few points extracted from my longer list of comments below, which I suggest the authors to discuss in their rebuttal. Feel free to also challenge the identified weaknesses above.
* L109: why consider the lazyfied version of the Markov kernel here to judge the performance of P? Do you actually recommend using P in practice?
* Proposition 2: what does it take for pi to be the unique stationary distribution?
* Figure 3: it seems that multiplying the number of trials by 10 decreases by 10 the number of iterations. So the gain comes from the parallelizability of MTM, right? Swapping sequential iterations for parallel ones is already interesting, but can you comment on how expensive parallelization is in your two examples (BVS and SBM)? Can you use linear algebraic tricks to further speed up parallel computation, as hinted in the text?

## Comments
* L60: at this stage of the paper, points 2 and 3 seem incompatible. Can you comment?
* L109: why consider the lazyfied version of the Markov kernel here to judge the performance of P? Do you actually recommend using P in practice?
* L122: this paragraph is slighlty confusing. Notions like "configuration of path network" are not defined. Also, from the notation L122, it is not clear whether Delta contains all paths from x to y, or is just a subset of these paths.
* L128ff: same for this paragraph. Can you give a textbook reference for this terminology, and maybe a figure in the appendix with a simple example?
* Theorem 1: can you give an idea of the proof? How do you build the right path-ensemble?
* L152: what do you mean by "applying the balancing rule"?
* Proposition 2: what does it take for pi to be the unique stationary distribution?
* L138: it would help to spend one paragraph going through one example (say, a simple case of BVS) and explain how the assumptions of Proposition 2 translate in that case.
* Section 4.1: what kernel do you use to infer other parameters, such as beta and phi? How does this impact the mixing time?
* Figure 3: it seems that multiplying the number of trials by 10 decreases by 10 the number of iterations. So the gain comes from the parallelizability of MTM, right? Swapping sequential iterations for parallel ones is already interesting, but can you comment on how expensive parallelization is in your two examples (BVS and SBM)? Can you use linear algebraic tricks to further speed up parallel computation, as hinted in the text?

## Typos
* L10 "improved performance over"
* L15 "MCMC methods have" or "The MCMC method has".
* L30 Ising model
* L72: I would say "Markov kernel" instead of "transition probability", but maybe this is just my habit.
* L130 rather "from transportation networks in graph theory"?
* L157 rather "Then, the Markov kernel P_{MTM} induced by Algorithm 1 has stationary pi".
* L172 rather "with number of variables p".
* L249 rather "until convergence of the Markov chain".
* L256 rather "the log probability ratio"
* L285 do you mean "wallclock time"?

**Ethics Review Area:**

["I don’t know"]

**Limitations:**

Adequately addressed.

**Strengths And Weaknesses:**

Overall, I like the paper, as it gives a thorough treatment of whether MTM improves the mixing time of MH. For a theoretical paper, it is well presented, although the text is sometimes a bit dense, and would benefit from having a running example like Bayesian variable selection, as well as a more detailed experimental section.

## Strengths
* a mixing time bound for MTM.
* a discussion and an algorithm for automatically choosing the number of trials.
* an honest treatment, showing that we gain in practice only if we can parallelize evaluations of the weight function.

## Weaknesses
* lack of details in the experimental section
* lack of comparison to other MCMC kernels for the same models, e.g. reversible-jump MCMC. However, since the main purpose of the paper is its theoretical bound and the improvement over classical MH, this is a small weakness.
* The improvement over classical MH seems to be swapping a factor in the number of sequential iterations for the same factor of parallelizable evaluations. Thus, I would expect more to be said on how efficiently the parallelization can be implemented, and how much can be gained in a challenging application on real data (only toy models are investigated).

---

> ### Author Response · Authors · 2022-08-01
> **Response to Reviewer 1 (Reviewer 8J7v) (part 1/2)**
>
> Thank you so much for the constructive review. Since there are 4 major overlapped comments raised by multiple reviewers, we created separate “Response to all” threads to address these common concerns. We also uploaded ‘**rebuttal.pdf**’ in the supplementary zip file, which contains all of our reviews, to present more detail with an  **easy-to-look presentation**.
>
>
> ### Strength and Weakness ###
>
> We feel truly grateful for your precise summary and positive comments.  Regarding the weakness, we could not include all the details of the experimental section in the main text due to the page limit, but we deferred them to the appendix. We have conducted two real data application analyses (see **Response to all: Section A** above) and comparative simulations with another MCMC method (see **Response to all: Section B** above), respectively. We also discuss parallelization in **Response to all: Section C**. Finally, we plan to add instructive illustrations and a running example of BVS to help understand the path method in revision.
>
> ### Questions and Comments ###
>
> Now we answer your questions and comments:
>
> > L109: why consider the lazyfied version of the Markov kernel here to judge the performance of P? Do you actually recommend using P in practice?
>
> **A:** We use $\mathbf{P}$ in practice. The reason we brought up $\mathbf{P}_{\mathrm{lazy}}$ is just for theoretical convenience. The lazy version is commonly used in theoretical works on the mixing times of Markov chains so that one does not need to worry about periodicity (and then one can bound the mixing time of a Markov chain using a bound on the spectral gap of the Markov transition matrix).
>
> > Proposition 2: what does it take for pi to be the unique stationary distribution?
>
> **A:** We elaborate on this in the first paragraph in Appendix A.1. Let $\mathbf{P}$ be the transition matrix of an arbitrary irreducible Markov chain on a finite space. Then there always exists a unique stationary probability distribution [Levin2017, Corollary 1.17]. Further, if the detailed balance condition is established for a probability distribution $\pi$ (that is, $\pi(x) \mathbf{P}(x,y) = \pi(y) \mathbf{P}(y,x)$ for all $x,y \in \mathcal{X}$), $\pi$ is the stationary distribution for $\mathbf{P}$. Assuming that our $\mathbf{P}$ is irreducible, what we need to show is whether $\pi$ and $\mathbf{P}_{\mathrm{MTM}}$ satisfy the detailed balance condition, which we showed in Appendix A.2.
>
> > Figure 3: it seems that multiplying the number of trials by 10 decreases by 10 the number of iterations. So the gain comes from the parallelizability of MTM, right? Swapping sequential iterations for parallel ones is already interesting, but can you comment on how expensive parallelization is in your two examples (BVS and SBM)? Can you use linear algebraic tricks to further speed up parallel computation, as hinted in the text?
>
> **A:** Please see "**Response to all: Section C**".
>
> > L60: at this stage of the paper, points 2 and 3 seem incompatible.
>
> **A:** We will revise point 2 to
>     "we prove that the mixing time bound of the MTM algorithm is smaller than that of the MH algorithm by a factor of the number of trials in a general model selection setting, under some regularity conditions including a rate condition on $N$."
>
> > L122: this paragraph is slighlty confusing. Notions like "configuration of path network" are not defined. Also, from the notation L122, it is not clear whether Delta contains all paths from x to y, or is just a subset of these paths.
>
> **A:** We used "configuration of path network'' to describe a path ensemble $\Delta$ in plain words, and we clarify that it is not a mathematical notation. In retrospect, it would have been less confusing if we quoted them.
>     $\Delta$ is built as follows. First, we pick only one path per $(x,y) \in \mathcal{X} \times \mathcal{X}$ (for $x \neq y$) and denote it as $\delta(x,y)$. Then, $\Delta$ is defined to be $\{\delta(x, y) \colon x, y \in \mathcal{X}, \, x \neq y \}$, which contains $|\mathcal{X}| \times (|\mathcal{X}| - 1)$ paths. We will make it clear in revision.
>
> > L128: Can you give a textbook reference for this terminology, and maybe a figure in the appendix with a simple example?
>
> **A:** A: Yes. We refer to [Sinclair1992], one of the seminal papers on path methods for finite Markov chains. For the textbook references, we recommend [Levin2017, Section 13.4] and [Saloff-Coste1997, Section 3]. We will also follow your suggestion to make a graphical illustration of path ensemble construction similar to Figure 3 in [Yang2016, Section 4.1] in the appendix in the revision.
>
>
> **(continued in part 2/2)**

---

> > ### Author Response · Authors · 2022-08-01
> > **Response to Reviewer 1 (Reviewer 8J7v) (part 2/2)**
> >
> >
> > **(continued from part 1/2)**
> > > Theorem 1: can you give an idea of the proof? How do you build the right path-ensemble?
> >
> > **A:** Yes. In a nutshell, the proof of Theorem 1 can be summarized as two key steps. The first step is to identify a suitable path ensemble $\Delta^*$. How we build $\Delta^*$ is described in line 64-84 of the appendix in detail. The brief idea of the construction is as follows. We first define a function $g$ on the state space $\mathcal{X}$ such that $g(x)$ is the highest posterior state among the predefined neighborhood of the state $x$. With the unimodality and finiteness of $\mathcal{X}$, some number of composition of $g$ for any state $x$ always leads to the mode $x^*$. By this fact, we can construct $\delta^*(x,y)$ for any $x \neq y$ by the rule described in line 79-84 in the appendix. The other major step is to bound the congestion parameter $\rho(\Delta^*)$. To this end, we need to control the lower bound of the transition probability $\mathbb{P}(x, g(x))$ from any state $x$ to $g(x)$. The proof of the lower bound is in line 93-123 of the appendix.
> >
> > > L152: what do you mean by "applying the balancing rule"?
> >
> > **A:** What we meant by the "balancing rule'' is the rule by a balancing function $h(u) = u h(1/u).$ To be more specific, once we define a function $h$ on $(0, 1]$, the function value $h(u)$ of $u \in (1, \infty)$ automatically defined by $h(u) = u h(1/u)$, which we referred to as the "balancing rule.'' We will revise it more explicitly without using the term "balancing rule.''
> >
> > > L138: it would help to spend one paragraph going through one example (say, a simple case of BVS) and explain how the assumptions of Proposition 2 translate in that case.
> >
> > **A:** In lines 159-200, we explained why the ordinary weight function can have a low acceptance probability and fail, which motivates the weight function (4) in Proposition 2. Especially, we used BVS as an example to illustrate its reason graphically (Figure 1 in the main text). Assuming your question was about Theorem 1 (since there is no assumption in Proposition 2), the argument in lines 159-200 for BVS is implicitly related to the assumptions in Theorem 1 (such as unimodality) and the path ensemble $\Delta^*$ (a red-marked local path $\gamma \rightarrow \gamma'$ in Figure 1).
> >     We will expand our discussion to more explicitly show how the assumptions of Theorem 1 translate with the BVS example in the revision.
> >
> > > Section 4.1: what kernel do you use to Failer other parameters, such as beta and phi? How does this impact the mixing time?
> >
> > **A:** For the BVS, since we are mainly interested in the binary indicator variable $\gamma$ so that $\beta$ and $\phi$ are marginalized out thanks to the conjugacy. That is, the target distribution of MTM sampler is $ \pi(\gamma|y) = \int \int \pi(\gamma,\beta,\phi | y)d\beta d\phi $ and draws of $\beta$ or $\phi$ does not affect the mixing time of $\gamma$. Same for the SBM where blockwise probabilities $Q_{uv}$ are marginalized out using the beta-binomial conjugacy.
> >
> > ### Typos ###
> >
> > For the typos, we deeply appreciate your corrections. We have thought about using "Markov kernel'' instead of "transition probability matrix'' since it is more general. However, since the scope of the manuscript is on the Markov chain with finite state space,
> > we would like to stick to "transition probability matrix'' to avoid potential confusion. Also, "wall time'' means wallclock time. We will fix the other typos in the revision. Thank you again!
> >
> > ### References ###
> >
> > [Levin2017] David A Levin and Yuval Peres. Markov chains and mixing times, volume 107. American Mathematical Soc., 2017.
> >
> > [Sinclair1992]  Alistair Sinclair. Improved bounds for mixing rates of Markov chains and multicommodity flow. Combinatorics, probability and Computing, 1(4):351–370, 1992
> >
> > [Saloff-Coste1997] Laurent Saloff-Coste. Lectures on finite Markov chains. Lectures on probability theory and statistics, pages 301–413, 1997.
> >
> > [Yang2016] Yun Yang, Martin J Wainwright, and Michael I Jordan. On the computational complexity of high-dimensional bayesian variable selection. The Annals of Statistics, 44(6):2497–2532, 2016.

---

> > > ### Comment · Reviewer_8J7v · 2022-08-04
> > > **Thanks**
> > >
> > > Thanks for the detailed answer!

---

### Author Response · Authors · 2022-08-01
**Response to all reviewers**

First of all, we deeply appreciate the three reviewers’ insightful and constructive comments. Before we provide point-by-point responses to the individual comments and questions, we would like to address four common concerns raised by multiple reviewers: **A. real data applications, B. comparison with other methods, C. parallelization,** and **D. state space of interest**, and will address them separately below in Section A, B, C, and D.

We also uploaded ‘**rebuttal.pdf**’ in the supplementary zip file, which contains all of our reviews, to present more detail with an  **easy-to-look presentation**.

---

> ### Author Response · Authors · 2022-08-01
> **Response to all: Section D. On the state space of interest**
>
>
> Our state space of interest is finite (so discrete), but the proposed locally balanced MTM algorithm is also applicable to continuous state spaces (which will be shown shortly). We choose to focus on the discrete case since the theory on continuous state spaces is usually developed under very different frameworks (and likewise, the theory on continuous spaces often cannot be readily applied to discrete ones).
> Indeed, developing MCMC theory or methodology on discrete spaces is often regarded as more challenging than on continuous ones [Zanella2020, Section 1], due to the lack of gradient information and a widely accepted theoretical framework supported by statistical theory (for comparison, on continuous spaces, one often assumes log-concavity or asymptotic normality of the target posterior distribution).
>
> To some extent, the proposed MTM method is conceptually similar to MALA (Metropolis adjusted Langevin algorithm) or HMC (Hamiltonian Monte Carlo) on continuous spaces in that MTM evaluates the "gradient'' by a random search of neighboring states. This suggests that for continuous-state-space problems where the gradient of log-posterior cannot be easily evaluated (e.g. Bayesian inverse problems and Gaussian process regression models), the proposed MTM method can be quite useful. Again, we choose to focus on model selection since it is already a very large and important class of statistical problems.
>
> We conclude this section with a simulation study that shows the weight function proposed in Proposition 2 can lead to an improved MTM algorithm on continuous spaces. Suppose our target distribution is the 10-dimensional Gaussian distribution $N(0, I_{10})$. We set our proposal kernel $q(\cdot | x ) = N_{10}(x, 10^{-2} I_{10})$, initialize the chains at $x_0 = (10, 10, \dots, 10)$, and run 10,000 iterations for each chain. The result is summarized in below Table, where for each setting we repeat the simulation 30 times. The advantage of the weight functions considered in Proposition 2 over $w_{\rm{ord}}$ is substantial. We present the log-posterior traceplots and the MCMC sample trajectories in **rebuttal.pdf**, Figures 2 and 3.
>
> *Table: Sampling from 10-d standard Gaussian distribution with 10,000 iterations using MTM. Averaged over 30 chains with random seed. The number in the parenthesis is the standard error*
> |                            |                     |                |                |                 |                |
> |----------------------------|---------------------|----------------|----------------|-----------------|----------------|
> |                            | $N$                 | 1     | 10             | 100             | 1,000          |
> | Acc. Rate | $w_{\mathrm{ord}}$  | 0.8604 (0.001) | 0.7488 (0.002) | 0.2971 (0.012)  | 0.0394 (0.008) |
> |                            | $w_{\mathrm{sqrt}}$ | $\cdot$        | 0.9667 (0.000) | 0.9897 (0.000)  | 0.9967 (0.000) |
> |                            | $w_{\mathrm{min}}$  | $\cdot$        | 0.9656 (0.000) | 0.9890 (0.000)  | 0.9963 (0.000) |
> |                            | $w_{\mathrm{max}}$  | $\cdot$        | 0.9593 (0.000) | 0.9861  (0.000) | 0.9950 (0.000) |
>
> ### Reference ###
>
> [Zanella2020] Informed proposals for local mcmc in discrete spaces. Journal of the American Statistical Association, 115(530):852–865, 2020.

---

> ### Author Response · Authors · 2022-08-01
> **Response to all: Section C. On parallelization**
>
> As we stated in lines 241-250, the overall theoretical computational complexity of MTM algorithm until the convergence remains the same as a usual MH algorithm. However, MTM enables parallel computations when evaluating $N$ weight functions and hence leads to a significant practical computational gain as evidenced by the reduced wall time reported in Table 1 in the main text.
> Under the random walk kernel $K_{RW}$, the evaluation of weight functions is equivalent to the evaluation of target distribution at $N$ states $y_1,\dots,y_N$. Here we briefly describe the computational strategy to simultaneously calculate $\pi(y_1),\dots,\pi(y_N)$ for BVS and SBM, where further details are going to be added in the revised manuscript. For BVS, since only one variable is added or deleted in the proposal, the Cholesky rank-1 update [Smith1996, George1997] is utilized to get $\pi(y_1),\dots,\pi(y_N)$ from $\pi(x)$. This calculation can be done in parallel by converting matrix-vector multiplication for calculating  $\pi(y_1)$ to matrix-matrix multiplication for calculating $\pi(y_1),\dots,\pi(y_N)$, and we refer Appendix B.1 of [Zanella2019] for details of this procedure. For SBM, let $A_i\in\\{0,1\\}^{p}$ be the $i$th column of adjacency matrix and $Z\in\\{0,1\\}^{p\times K}$ be one-hot encoded partition matrix such that  $Z_{i,k}=1$ if node $i$ is in cluster $k$ and 0 otherwise. Since a node is assigned to another cluster one at a time, the calculation of $\pi(y_1)\dots,\pi(y_N)$ given current state $\pi(x)$ can be done by counting the change of the number of edges between clusters; see eq.(5) of [Zhuo2021]. Letting $A_J\in\\{0,1\\}^{p\times N}$ where column $A_j$ corresponds to the $j$th proposal, the matrix-matrix multiplication $Z^\top A_J$ allows to calculate $\pi(y_1)\dots,\pi(y_N)$ simultaneously from $\pi(x)$. In addition, if the graph is sparse, then sparse matrix multiplication algorithms can be utilized for further speedup. We believe for the complex real data application scenarios where the target distribution $\pi(\cdot)$ is very expensive to evaluate, the benefit of parallel computation of multiple trials is even more prominent.
>
> ### References ###
>
> [Smith1996] Michael Smith and Robert Kohn. Nonparametric regression using Bayesian variable selection. Journal of Econometrics, 75(2):317–343, 1996.
>
> [George1997] Edward I George and Robert E McCulloch. Approaches for bayesian variable selection.Statistica sinica, pages 339–373, 1997
>
> [Zanella2019] Giacomo Zanella and Gareth Roberts. Scalable importance tempering and Bayesian variable selection. Journal of the Royal Statistical Society: Series B (Statistical Methodology), 81 (3):489–517, 2019.
>
> [Zhuo2021] Bumeng Zhuo and Chao Gao. Mixing time of Metropolis-Hastings for Bayesian community detection. J. Mach. Learn. Res., 22:10–1, 2021

---

> ### Author Response · Authors · 2022-08-01
> **Response to all: Section B. Additional comparison methods**
>
> Although the main focus of the paper is to theoretically study the mixing time for the family of MTM algorithms and hence the main goal of our experiments in Section 4 is to empirically verify our theoretical insights (that the MTM mixing time is smaller by a factor of $N$ and that a balancing weight function tends to perform better), we agree with reviewers that the comparison with other methods can be very useful to readers.
>
> **RJ-MCMC.** Reviewers suggested that we compare MTM with reversible-jump MCMC methods [Green1995]. We clarify that the MTM algorithms used in both BVS and spatial clustering examples fall within the family of RJMCMC but model parameters ($\beta$ and $\phi$ for BVS, $\mu$ and $\sigma^2$ for spatial clustering) are marginalized out.
>
> **Locally balanced MH.** ([Zanella2020])
> We choose to add a comparison with LB because: (1) it is a very recently developed locally informed MH algorithm also for discrete spaces, and (2) it has been reported [Zanella2020] to outperform the other state-of-the-art methods.
> We compare the performance of MTM with LB under the same setting as described in Section 4. We run LB with 3 choices of balancing function $\sqrt{t}, \min\{1,t\}, \max\{1,t\}$ and report its results in the last column of the following table, which extends the result of Table 1 in the main text. For the comparison results of other simulation settings, please see "**Additional tables**" section in "**rebuttal.pdf**".
>
> *Table 3: Summary of $T_H$, median wall time (in seconds) until the Markov chain hit the highest posterior state over 50 replicates. Last column corresponds to the locally balanced MH algorithm of [Zanella2020], where ``Fail'' indicates that chains never hit the highest posterior state in more than half of the replicated datasets.*
>
> | $T_H$                                      | $N$                 | 5    | 10            | 50   | 100  | 500           | 1000 | 2000 | 5000  | LB   |
> |--------------------------------------------|---------------------|------|---------------|------|------|---------------|------|------|-------|------|
> | BVS, indep, SNR = 4       | $w_{\mathrm{sqrt}}$ | 0.81 | 0.46          | 0.11 | 0.07 | **0.04** | 0.05 | 0.09 | 0.30  | Fail |
> |                                            | $w_{\mathrm{min}}$  | 0.89 | 0.53          | 0.12 | 0.07 | **0.04** | 0.04 | 0.06 | 0.11  | 0.07 |
> |                                            | $w_{\mathrm{max}}$  | 0.88 | 0.53          | 0.13 | 0.07 | **0.04** | 0.05 | 0.09 | 0.33  | Fail |
> | SBM, $K=2$, $CH\approx 2$ | $w_{\mathrm{sqrt}}$ | 0.4  | **0.35** | 0.48 | 0.75 | 1.49          | 2.15 | 3.78 | 9.30  | 1.28 |
> |                                            | $w_{\mathrm{min}}$  | 0.56 | **0.41** | 0.57 | 0.89 | 1.55          | 2.62 | 4.31 | 11.26 | 1.59 |
> |                                            | $w_{\mathrm{max}}$  | 0.47 | **0.39** | 0.51 | 0.75 | 1.29          | 2.07 | 3.57 | 8.35  | 1.35 |
>
> In terms of hitting time $T_H$, LB is not as efficient as MTM with a smaller choice of $N$. Especially in high signal case, LB with balancing functions $\sqrt{t}$ and $\max\{1,t\}$ is often stuck at the local mode. It is easier for MTM to escape from such local modes by randomly searching part of its neighborhood to select the proposal.
>
> ### References ###
>
> [Green1995] Peter J Green. Reversible jump Markov chain Monte Carlo computation and Bayesian model determination. Biometrika, 82(4):711–732, 1995.
>
> [Zanella2020] Giacomo Zanella. Informed proposals for local MCMC in discrete spaces. Journal of the American Statistical Association, 115(530):852–865, 2020.

---

> ### Author Response · Authors · 2022-08-01
> **Response to all: Section A. Real data examples**
>
> ### A.1. Real data example 1: GWAS dataset for Bayesian variable selection ###
>
> We consider a genome-wide association study (GWAS) dataset on glaucoma studied in [Zhou2022] with sample size $n=5418$ and number of genetic variants $p=7255$.  The response variable $\mathbf{y}\in\mathbb{R}^{5418}$ is the standardized cut-to-disk ratio measurements averaged over two eyes. We use the BVS model described in Section 4.1, with hyperparameters $\mathscr{G} = 100$ and $\kappa = 0.8$. Since the "true" state is not available, we compare the acceptance rate and the number of unique states visited, averaged over 5 chains. From the table below, it is clear that the performance of $w_{ord}$ deteriorates significantly as $N$ grows whereas $w_{sqrt}$, $w_{min}$, $w_{max}$ does not.
>
> *Table: GWAS dataset analysis results using MTM, averaged over 5 chains with random seeds*
> |         |      |        |                |        |                |                |        |
> |-------|-------|--------|----------------|--------|----------------|----------------|--------|
> |     | $N$                 | 50     | 100            | 500    | 1000           | 2000           | 5000   |
> |     | iteration           | $10^6$ | $5\times 10^5$ | $10^5$ | $5\times 10^4$ | $2\times 10^4$ | $10^4$ |
> | Acc. Rate   | $w_{ord}$  | 0.4014 | 0.5081         | 0.1370 | 0.0471         | 0.0259         | 0.0085 |
> |     | $w_{sqrt}$ | 0.3407 | 0.4812         | 0.7571 | 0.8325         | 0.8777         | 0.9251 |
> |     | $w_{min}$  | 0.4136 | 0.5851         | 0.8252 | 0.8797         | 0.9138         | 0.9455 |
> |     | $w_{ax}$  | 0.2335 | 0.3404         | 0.6161 | 0.7199         | 0.7930         | 0.8698 |
> | Number of unique states visited | $w_{ord}$  | 199442 | 126238         | 6796   | 1172           | 259            | 43     |
> |      | $w_{sqrt}$ | 169242 | 119396         | 37481  | 20588          | 8669           | 4563   |
> |     | $w_{min}$  | 205459 | 145372         | 40993  | 21848          | 9080           | 4696   |
> |   | $w_{ax}$  | 115615 | 84124          | 30227  | 17591          | 7717           | 4211   |
>
> We also report the posterior inclusion probabilities of the top 10 genetic variants in **rebuttal.pdf** in Tables 5-10. All results generally agree with the result of [Zhou2022], except that when we use $w_{ord}$ with $N=5000$, the chain is stuck at local modes and fails to find the significant genetic variants.
>
> ### A.2 Real data example 2: Single-cell RNA dataset for structure learning  ###
>
> We consider a gene expression dataset on Alzheimer's disease used in [Chang2022] with sample size $n=1,666$ and number of genes $p=73$. The goal is to learn the underlying directed acyclic graph (DAG) model among the $p$ genes.
> Due to acyclicity, each DAG has at least one ordering of the nodes.
> For example, the ordering for the DAG $a \rightarrow b \leftarrow c$ can be either $(a, c, b)$ or $(c, a, b)$.
> A popular Bayesian structure learning strategy is to use MCMC sampling to first learn the marginal posterior distribution on the order space and then find one or multiple best DAGs for each sampled ordering.
>
> We use an MTM implementation of the order MCMC sampler proposed in [Chang2022], which aims to learn the posterior distribution on the order space $\mathbb{S}^p$, the permutation group on $\{1, \dots, p\}$.
> For $p= 73$, the size of our model space $\mathbb{S}^p$ is equal to $73! \approx 4.5 \times 10^{105}$.
> For each weight function and each setting, we simulate 30 chains, initialized at $(1,\dots, 73)$.
> It is clear from the below table that the acceptance probability with ordinary weight function $w_{ord}$ significantly deteriorates, which is consistent with our theory. We can see this tendency more clearly in the log-posterior traceplots for all weight functions in **rebuttal.pdf**, Figure 1
>
> *Table: The single-cell RNA database for Alzheimer’s disease. Averaged over 30 chains with random seeds.
> The number in the parenthesis is the standard error.*
> |   |    |     |       |
> |------|------|-----------|---------|
> |    | $N$    | 5    | 50   |
> |  | iteration   | $5\times 10^2$ | $2 \times 10^2$ |
> | Acc. Rate    | $w_{ord}$  | 0.7187 (0.004) | 0.0012 (0.000)  |
> | | $w_{sqrt}$ | 0.8029 (0.004) | 0.9186 (0.003)  |
> | | $w_{min}$  | 0.8329 (0.002) | 0.9506 (0.001)  |
> | | $w_{ax}$  | 0.6643 (0.006) | 0.6806 (0.008)  |
> | Number of unique states visited | $w_{ord}$  | 361.1 (3.1)    | 1.333333 (0.1)  |
> | | $w_{sqrt}$ | 402.3 (2.2)    | 184.3 (0.8)     |
> | | $w_{min}$  | 416.7 (1.8)    | 191.8 (0.6)     |
> | | $w_{ax}$  | 332.0 (2.9)    | 136.2 (1.6)     |
>
> ### References ###
>
> [Chang2022] Hyunwoong Chang, James Cai, and Quan Zhou. Order-based structure learning without score equivalence. arXiv preprint arXiv:2202.05150, 2022
>
> [Zhou2022] Quan Zhou, Jun Yang, Dootika Vats, Gareth O Roberts, and Jeffrey S Rosenthal. Dimension-free mixing for high-dimensional bayesian variable selection. Journal of the Royal Statistical Society: Series B, to appear., 2022+

---

### Meta-Review · Area_Chair_6sgo · 2022-08-22

**Recommendation:** Accept
**Confidence:** Certain

**Metareview:**

Based on the reviews and discussions, we are happy to recommend acceptance. Please make sure that all comments in the discussion threads are taken into account in the final version of the manuscript.


**Award:**

No

---

### Decision · Program_Chairs · 2022-09-14

Accept